# HIERARCHICAL GAUSSIAN MIXTURE BASED TASK GENERATIVE MODEL FOR ROBUST META-LEARNING

## ABSTRACT

Meta-learning enables quick adaptation of machine learning models to new tasks with limited data. While tasks could come from varying distributions in reality, most of the existing meta-learning methods consider both training and testing tasks as from the same uni-component distribution, overlooking two critical needs of a practical solution: (1) the various sources of tasks may compose a multi-component mixture distribution, and (2) novel tasks may come from a distribution that is unseen during meta-training. In this paper, we demonstrate these two challenges can be solved jointly by modeling the density of task instances. We develop a meta-training framework underlain by a novel Hierarchical Gaussian Mixture based Task Generative Model (HTGM). HTGM extends the widely used empirical process of sampling tasks to a theoretical model, which learns task embeddings, fits the mixture distribution of tasks, and enables density-based scoring of novel tasks. The framework is agnostic to the encoder and scales well with large backbone networks. The model parameters are learned end-to-end by maximum likelihood estimation via an Expectation-Maximization algorithm. Extensive experiments on benchmark datasets indicate the effectiveness of our method for both sample classification and novel task detection.

## 1 INTRODUCTION

Training models in small data regimes is of fundamental importance. It demands a model's ability to quickly adapt to new environments and tasks. To compensate for the lack of training data for each task, meta-learning (*a.k.a.* learning to learn) has become an essential paradigm for model training by generalizing meta-knowledge across tasks (Snell et al., 2017; Finn et al., 2017). While most existing meta-learning approaches were built upon an assumption that all training/testing tasks are sampled from the same distribution, a more realistic scenario should accommodate training tasks that lie in a mixture of distributions, and testing tasks that may belong to or deviate from the learned distributions. For example, in recent medical research, a global model is typically trained on the historical medical records of a certain set of patients in the database (Shukla & Marlin, 2019; Wu et al., 2021). However, due to the uniqueness of individuals (*e.g.*, gender, age, genetics), patients' data have a substantial discrepancy, and the pre-trained model may demonstrate significant demographic or geographical biases when testing on a new patient (Purushotham et al., 2017). This issue can be mitigated by personalized medicine approaches (Chan & Ginsburg, 2011; Ni et al., 2022) where each patient is regarded as a task, and the pre-trained model is fine-tuned (*i.e.*, personalized) on a support set of a few records collected in a short period (*e.g.*, a few weeks) from every patient for adaptation. In this case, the training tasks (*i.e.*, patients) could be sampled from a mixture of distributions (*e.g.*, different age groups), and a testing task may or may not belong to any of the observed groups. As such, a meta-training strategy that is able to fit a mixture of task distributions and identify novel tasks is desirable for making meta-learning a practical solution.

One way to tackle the mixture distributions of tasks is to tailor the transferable knowledge to each task by learning a task-specific representation (Oreshkin et al., 2018; Vuorio et al., 2018; Lee & Choi, 2018), but as discussed in (Yao et al., 2019a), the over-customized knowledge prevents its generalization among closely related tasks (*e.g.*, tasks from the same distribution). The more recent methods try to balance the generalization and customization of the meta-knowledge by promoting *local generalization* either among a cluster of related tasks (Yao et al., 2019a), or within a neighborhood of a meta-knowledge graph of tasks (Yao et al., 2019b). Neither of them explicitly learns the

underlying distribution from which the tasks are generated, rendering them infeasible for detecting novel tasks that are out-of-distribution. However, detecting novel tasks is crucial in high-stake domains, such as medicine and finance, which provides users (*e.g.*, physicians) confidence on whether to trust the results of a testing task or not, and facilitates the downstream decision-making.

In (Lee et al., 2019a), a task-specific tuning variable was introduced to modulate the initial parameters learned by MAML (Finn et al., 2017), so that the impacts of the meta-knowledge on different tasks are adjusted differently, *e.g.*, novel tasks receive less impact than known tasks do. Whereas, this method focuses on improving model performance on different tasks (either known or novel), but neglects the critical mission of detecting which tasks are novel. In practice, providing an unreliable accuracy on a novel task, without differentiating it from other tasks may be meaningless and risky.

Since the aforementioned methods cannot simultaneously handle the mixture distribution of tasks and novel tasks, a practical solution is in demand. In this work, we consider tasks as instances, and demonstrate the dual problem of modeling the mixture of task distributions and detecting novel tasks are two sides of the same coin, *i.e.,* density estimation on task instances. To this end, we propose a new Hierarchical Gaussian Mixture based Task Generative Model (HTGM) to explicitly model the generative process of task instances. Our contributions are summarized as follows.

- For the first time, the widely used empirical process of generating a task is theoretically extended to and specified by a hierarchy of Gaussian mixture (GM) distributions. HTGM generates a *task embedding* from a *task-level* GM, and uses it to define the task-conditioned mixture probabilities for a *class-level* GM, from which samples are drawn, for instantiating the generated task. To allow realistic classes per task, a new Gibbs distribution is proposed to underlie the class-level GM.

- HTGM is an encoder-agnostic framework, thus is flexible to different domains. It inherits metric-based meta-learning methods, and only introduces a small overhead to an encoder for parameterizing its distributions, thus is efficient, and enables large-scale backbone networks. The model parameters are learned end-to-end by maximum likelihood estimation via a principled Expectation-Maximization (EM) algorithm. The bounds of our likelihood function is theoretically analyzed.

- In the experiments, we evaluated HTGM on benchmark image datasets for validating its ability to take advantage of large backbone networks, its effectiveness in modeling the mixture distribution of tasks, and its usefulness in identifying novel tasks. The results demonstrate HTGM outperforms the state-of-the-art (SOTA) baselines with significant improvements in most cases.

## 2 RELATED WORK

To the best of our knowledge, this is the first work to explicitly model the generative process of task instances from a mixture of distributions for meta-learning with novel task detection. Meta-learning aims to handle the few-shot learning problem, which derives memory-based (Mishra et al., 2018), optimization-based (Finn et al., 2017; Li et al., 2017), and metric-based methods (Vinyals et al., 2016; Snell et al., 2017), which often consider an artificial scenario where training/test tasks are sampled from the same distribution. To enable more varying tasks, task-adaptive methods facilitates the customization of meta-knowledge by learning task-specific parameters (Rusu et al., 2018; Lee & Choi, 2018), temperature scaling parameters (Oreshkin et al., 2018), and task-specific modulation on model initialization (Vuorio et al., 2018; Yao et al., 2019a;b; Lee et al., 2019a). Among them, there are methods tackling the mixture distribution of tasks by clustering tasks (Yao et al., 2019a) or learning task graphs (Yao et al., 2019b), and method relocating the initial parameters for different tasks so that they use the meta-knowledge differently (Lee et al., 2019a). As discussed before, none of these methods jointly handle the mixture of task distributions and the detection of novel tasks.

Our model is built upon metric-based methods, and learns task embeddings for modeling task distributions. Achille et al. (2019) also proposed to learn embeddings for tasks and introduced a meta-learning method, but not for few-shot learning. Its embeddings are from a pre-specified set of tasks (rather than episode-wise sampling), and the meta-learning framework is for model selection. The model in (Yao et al., 2019a) has an augmented encoder for task embedding, but it does not explicitly model task generation, and is not designed for novel task detection (empirical comparison in 4.1).

Conventional novelty detection aims to identify and reject samples from unseen classes (Cheng & Vasconcelos, 2021). It relates to open-set recognition (Vaze et al., 2022), which aims to simultaneously identify unknown samples and classify samples from known classes. Out-of-distribution

(OOD) detection (Liang et al., 2018; Liu et al., 2020) can be seen as a special case of novelty detection where novel samples are from other problem domains or datasets, thus are considered to be easier to detect than novelties (Cheng & Vasconcelos, 2021). These methods are for large-scale training. In contrast, we want to detect novel tasks, which is a new problem in the small data regime.

Hierarchical Gaussian Mixture (HGM) model has appeared in some traditional works (Goldberger & Roweis, 2005; Olech & Paradowski, 2016; Athey et al., 2019) for hierarchical clustering by applying GM agglomeratively or divisively, which do not pre-train models for meta-learning, and is remarkably different from the topic in this paper. The differences are elaborated in Appendix B.1. Moreover, we discuss the relevant multi-task learning methods with task grouping in Appendix B.2.

## 3 HIERARCHICAL GAUSSIAN MIXTURE BASED TASK GENERATIVE MODEL

Meta-learning methods typically use an *episodic learning* strategy, where the meta-training set $\mathcal{D}^{tr}$ consists of a batch of episodes. Each episode samples a task $\tau$ from a distribution $p(\tau)$. Task $\tau$ has a support set $\mathcal{D}^s_\tau = \{(\mathbf{x}^s_i, y^s_i)\}^{n_s}_{i=1}$ for training, and a query set $\mathcal{D}^q_\tau = \{(\mathbf{x}^q_i, y^q_i)\}^{n_q}_{i=1}$ for testing, where $n_s$ is a small number to denote a few training samples. In particular, in a commonly used $N$-way $K$-shot $Q$-query task (Vinyals et al., 2016), $\mathcal{D}^s_\tau$ and $\mathcal{D}^q_\tau$ contain $N$ classes, with $K$ and $Q$ samples per class respectively, *i.e.*, $n_s = NK$ and $n_q = NQ$.

Let $f_{\boldsymbol{\theta}}(\mathbf{x}^*_i) \to y^*_i$ be a base model ($*$ denotes s or q), and $f_{\boldsymbol{\theta}}(\cdot; \mathcal{D}^s_\tau)$ be the adapted model on $\mathcal{D}^s_\tau$. The training objective on $\tau$ is to minimize the average test error of the adapted model, *i.e.*, $\mathbb{E}_{(\mathbf{x}^q_i, y^q_i) \in \mathcal{D}^q_\tau} \ell(y^q_i, f_{\boldsymbol{\theta}}(\mathbf{x}^q_i; \mathcal{D}^s_\tau))$, where $\ell(\cdot, \cdot)$ is a loss function (*e.g.*, cross-entropy loss), and the meta-training process aims to find the parameter $\boldsymbol{\theta}$ that minimizes this error over all episodes in $\mathcal{D}^{tr}$. Then, $f_{\boldsymbol{\theta}}$ is evaluated on every episode of a meta-test set $\mathcal{D}^{te}$ that samples a task from the same distribution $p(\tau)$. Usually, $p(\tau)$ is a simple distribution (Finn et al., 2017; Lee et al., 2019a). In this work, $p(\tau)$ is generalized to a mixture distribution consisting of multiple components $p_1(\tau), ..., p_r(\tau)$, and a test episode may sample a task either in or out of any component of $p(\tau)$. As such, given the training tasks in $\mathcal{D}^{tr}$, our goal is to estimate the underlying density of $p(\tau)$, so that once a test task is given, we can (1) identify if it is a novel task, and (2) adapt $f_{\boldsymbol{\theta}}$ to it with optimal accuracy.

Specifically, the base model $f_{\boldsymbol{\theta}}$ can be written as a combination of an encoder $g_{\boldsymbol{\theta}_e}$ and a predictor $h_{\boldsymbol{\theta}_p}$, *i.e.*, $f_{\boldsymbol{\theta}}(\mathbf{x}^*_i) = h_{\boldsymbol{\theta}_p}(g_{\boldsymbol{\theta}_e}(\mathbf{x}^*_i))$ (Tian et al., 2020). In this work, we focus on a metric-based non-parametric learner, *i.e.*, $\boldsymbol{\theta}_p = \varnothing$ (*e.g.*, prototypical networks (Snell et al., 2017)), not only because metric-based classifiers were confirmed as more effective than probabilistic classifiers for novelty detection (Jeong et al., 2021), but also for its better training efficiency that fits large-scale backbone networks than the costly nested-loop training of optimization-based methods (Tian et al., 2020).

Formally, our goal is to find the model parameter $\boldsymbol{\theta}$ that maximizes the likelihood of observing a task $\tau$. In other words, let $f_{\boldsymbol{\theta}}(\mathbf{x}^*_i) = \mathbf{e}^*_i \in \mathbb{R}^d$ be the sample embedding, we want to maximize the likelihood of the joint distribution $p_{\boldsymbol{\theta}}(\mathbf{e}^*_i, y^*_i)$ on the observed data in $\mathcal{D}_\tau = \{\mathcal{D}^s_\tau, \mathcal{D}^q_\tau\}$. We consider each task $\tau$ as an instance, with a representation $\mathbf{v}_\tau \in \mathbb{R}^d$ in the embedding space (the method to infer $\mathbf{v}_\tau$ is described in Sec. 3.2). To model the unobserved mixture component, we associate every task with a latent variable $z_\tau$ to indicate to which component it belongs. Suppose there are $r$ possible components, and let $n = n_s + n_q$ be the total number of samples in $\mathcal{D}_\tau$, the log-likelihood to maximize can be written by hierarchically factorizing it on $y^*_i$ and marginalizing out $\mathbf{v}_\tau$ and $z_\tau$.

$$
\begin{aligned}
\ell(\mathcal{D}_\tau; \boldsymbol{\theta}) &= \frac{1}{n} \sum_{i=1}^{n} \log \left[ p_{\boldsymbol{\theta}}(\mathbf{e}^*_i, y^*_i) \right] = \frac{1}{n} \sum_{i=1}^{n} \log \left[ p_{\boldsymbol{\theta}}(\mathbf{e}^*_i | y^*_i) p(y^*_i) \right] \\
&= \frac{1}{n} \sum_{i=1}^{n} \log \left[ p_{\boldsymbol{\theta}}(\mathbf{e}^*_i | y^*_i) \left[ \int_{\mathbf{v}_\tau} p(y^*_i | \mathbf{v}_\tau) p(\mathbf{v}_\tau) d\mathbf{v}_\tau \right] \right] \\
&= \frac{1}{n} \sum_{i=1}^{n} \log \left[ p_{\boldsymbol{\theta}}(\mathbf{e}^*_i | y^*_i) \left[ \int_{\mathbf{v}_\tau} p(y^*_i | \mathbf{v}_\tau) \left[ \sum_{z_\tau=1}^{r} p(\mathbf{v}_\tau | z_\tau) p(z_\tau) \right] d\mathbf{v}_\tau \right] \right]
\end{aligned}
\tag{1}
$$

where $p_{\boldsymbol{\theta}}(\mathbf{e}^*_i | y^*_i)$ specifies the probability of sampling $\mathbf{e}^*_i$ from the $y^*_i$-th class, $p(y^*_i | \mathbf{v}_\tau)$ is the probability of sampling the $y^*_i$-th class for task $\tau$, and $p(\mathbf{v}_\tau | z_\tau)$ indicates the probability of generating a task $\tau$ from the $z_\tau$-th mixture component. $p(z_\tau)$ is a prior on the $z_\tau$-th component. Hence, Eq. (1) implies a generative process of task $\tau$: $z_\tau \to \mathbf{v}_\tau \to y^*_i \to \mathbf{e}^*_i$. Next, we define each of the aforementioned distributions and propose our HTGM method.

### 3.1 MODEL SPECIFICATION AND PARAMETERIZATION

In Eq. (1), the *class-conditional distribution* $p_{\boldsymbol{\theta}}(\mathbf{e}_i^*|y_i^*)$, the *task-conditional distribution* $p(y_i^*|\mathbf{v}_\tau)$, and the *mixture distribution of tasks* defined by $\{p(\mathbf{v}_\tau|z_\tau), p(z_\tau)\}$ are not specified. To make Eq. (1) optimizable, we introduce our HTGM that models the generative process of tasks. Because $\mathcal{D}_\tau^{\mathsf{s}}$ and $\mathcal{D}_\tau^{\mathsf{q}}$ follow the same distribution, in the following, we ignore the superscript $*$ for simplicity.

**Class-Conditional Distribution.** First, similar to (Lee et al., 2018; 2019b), we use Gaussian distribution to model the embeddings $\mathbf{e}_i$'s in every class. Let $\boldsymbol{\mu}_{y_i}^{\mathsf{c}}$ and $\boldsymbol{\Sigma}_{y_i}^{\mathsf{c}}$ be the mean and variance of the distribution of the $y_i$-th class, then $p_{\boldsymbol{\theta}}(\mathbf{e}_i|y_i) = \mathcal{N}(\mathbf{e}_i|\boldsymbol{\mu}_{y_i}^{\mathsf{c}}, \boldsymbol{\Sigma}_{y_i}^{\mathsf{c}})$. In fact, the samples in all of the classes of task $\tau$ comprise a Gaussian mixture distribution, where $p(y_i)$ is the *mixture probability* of the $y_i$-th class. In Eq. (1), $p(y_i)$ is factorized to be task-specific, *i.e.*, $p(y_i|\mathbf{v}_\tau)$, which resorts to another mixture distribution $p(\mathbf{v}_\tau)$ of tasks, and establishes a structure of hierarchical mixture.

**Task-Conditional Distribution.** A straightforward definition of $p(y_i|\mathbf{v}_\tau)$ is the density at $\boldsymbol{\mu}_{y_i}^{\mathsf{c}}$ in a Gaussian distribution with $\mathbf{v}_\tau$ as the mean, where $\boldsymbol{\mu}_{y_i}^{\mathsf{c}}$ is the mean (or prototype) of the $y_i$-th class. However, doing so exposes two problems: (1) the density function of Gaussian distribution is log-concave with one global maximum. Given the mean and variance, maximizing its log-likelihood tends to collapse the prototypes $\boldsymbol{\mu}_{y_i}^{\mathsf{c}}$'s of all classes in $\tau$, making them indistinguishable and impairing classification; (2) given $\mathbf{v}_\tau$, this method tends to sample classes with small $D_{\mathbf{v}_\tau}(\boldsymbol{\mu}_{y_i}^{\mathsf{c}})$, where $D_{\mathbf{v}_\tau}(\cdot)$ measures the Mahalanobis distance between a data point and the Gaussian distribution centered at $\mathbf{v}_\tau$. However, in most of the existing works, classes are often uniformly sampled from a domain without any prior on distances (Finn et al., 2017). Fitting the distance function with such "uniform" classes naively leads to an ill-posed learning problem with degenerated solutions. In light of these issues, we seek to define $p(y_i|\mathbf{v}_\tau)$ as a (parameterized) density function with at least $N$ global optimums so that it can distinguish the $N$ different class prototypes of $N$-way tasks. The $N$ equal (global) optimums also allow it to fit $N$ classes uniformly sampled from a domain. To this end, let $\boldsymbol{\mu}_k^{\mathsf{c}}$ be the *surrogate embedding* of the $k$-th class, we propose a Gibbs distribution $\pi(\boldsymbol{\mu}_k^{\mathsf{c}}|\mathbf{v}_\tau, \boldsymbol{\omega})$ defined by $\mathbf{v}_\tau$ and trainable parameters $\boldsymbol{\omega}$ with an energy function. Then we write $p(y_i = k|\mathbf{v}_\tau)$ as

$$p_{\boldsymbol{\omega}}(y_i = k|\mathbf{v}_\tau) = \pi(\boldsymbol{\mu}_k^{\mathsf{c}}|\mathbf{v}_\tau, \boldsymbol{\omega}) = \frac{\exp\left[-E_{\boldsymbol{\omega}}(\boldsymbol{\mu}_k^{\mathsf{c}}; \mathbf{v}_\tau)\right]}{\int_{\boldsymbol{\mu}_k^{\mathsf{c}}} \exp\left[-E_{\boldsymbol{\omega}}(\boldsymbol{\mu}_k^{\mathsf{c}}; \mathbf{v}_\tau)\right]} \tag{2}$$

where $E_{\boldsymbol{\omega}}(\boldsymbol{\mu}_k^{\mathsf{c}}; \mathbf{v}_\tau) = \min\left(\{||\boldsymbol{\mu}_k^{\mathsf{c}} - \mathbf{W}_j\mathbf{v}_\tau||_2^2\}_{j=1}^N\right)$ is our energy function, and the denominator in Eq (2) is a normalizing constant (with respect to $\boldsymbol{\mu}_k^{\mathsf{c}}$), *a.k.a.* the partition function in an energy-based model (EBM) (LeCun et al., 2006). $\boldsymbol{\omega} = \{\mathbf{W}_1, ..., \mathbf{W}_N\}$ are trainable parameters, with $\mathbf{W}_i \in \mathbb{R}^{d \times d}$. Given $\boldsymbol{\omega}$ and $\mathbf{v}_\tau$, Eq. (2) has $N$ global maximums at $\boldsymbol{\mu}_k^{\mathsf{c}} = \mathbf{W}_1\mathbf{v}_\tau, ..., \boldsymbol{\mu}_k^{\mathsf{c}} = \mathbf{W}_N\mathbf{v}_\tau$. More interpretations of the proposed task-conditional distribution can be found in Appendix B.3.

**Mixture Distribution of Tasks.** In Eq. (1), the task distribution $p(\mathbf{v}_\tau)$ is factorized as a mixture of $p(\mathbf{v}_\tau|z_\tau = 1), ..., p(\mathbf{v}_\tau|z_\tau = r)$, weighted by their respective mixture probability $p(z_\tau)$. Thus we specify $p(\mathbf{v}_\tau)$ as a Gaussian mixture distribution, and introduce $\boldsymbol{\mu}_{z_\tau}^{\mathsf{t}}$ and $\boldsymbol{\Sigma}_{z_\tau}^{\mathsf{t}}$ as the mean and variance for each component, *i.e.*, $p(\mathbf{v}_\tau|z_\tau) = \mathcal{N}(\mathbf{v}_\tau|\boldsymbol{\mu}_{z_\tau}^{\mathsf{t}}, \boldsymbol{\Sigma}_{z_\tau}^{\mathsf{t}})$. Then the generation of $\mathbf{v}_\tau$ involves two steps: (1) draw a latent variable $z_\tau$ from a categorical distribution on $[p(z_\tau = 1), ..., p(z_\tau = r)]$, which can be Uniform($r$), and (2) draw $\mathbf{v}_\tau$ from $\mathcal{N}(\boldsymbol{\mu}_{z_\tau}^{\mathsf{t}}, \boldsymbol{\Sigma}_{z_\tau}^{\mathsf{t}})$ (Bishop, 2006).

As such, our HTGM generative process of an $N$-way $K$-shot $Q$-query task $\tau$ can be summarized as

1. Draw a latent task variable $z_\tau \sim \text{Categorical}([p(z_\tau = 1), ..., p(z_\tau = r)])$
2. Draw a task embedding $\mathbf{v}_\tau \sim \mathcal{N}(\boldsymbol{\mu}_{z_\tau}^{\mathsf{t}}, \boldsymbol{\Sigma}_{z_\tau}^{\mathsf{t}})$
3. For $k = 1, ..., N$:
   - (a) Draw a class prototype $\boldsymbol{\mu}_k^{\mathsf{c}} \sim \pi(\boldsymbol{\mu}_k^{\mathsf{c}}|\mathbf{v}_\tau, \boldsymbol{\omega})$ from the proposed Gibbs distribution in Eq. (2)
   - (b) For $i = 1, ..., K + Q$:
     - i. Set $y_i = k$, draw a sample embedding $\mathbf{e}_i \sim \mathcal{N}(\mathbf{e}_i|\boldsymbol{\mu}_{y_i}^{\mathsf{c}}, \boldsymbol{\Sigma}_{y_i}^{\mathsf{c}})$
     - ii. Allocate $(\mathbf{e}_i, y_i)$ to the support set $\mathcal{D}_\tau^{\mathsf{s}}$ if $i \leq K$; else allocate $(\mathbf{e}_i, y_i)$ to the query set $\mathcal{D}_\tau^{\mathsf{q}}$

To reduce complexity, we investigate the feasibility of using isotropic Gaussian with tied variance, *i.e.*, $\boldsymbol{\Sigma}_1^{\mathsf{c}} = ... = \boldsymbol{\Sigma}_N^{\mathsf{c}} = \sigma^2\mathbf{I}$, for class distributions, which turned out to be efficient in our experiments. Here, $\mathbf{I}$ is the identity matrix, $\sigma$ is a hyperparameter. Tied variance is also a commonly used trick in Gaussian discriminate analysis (GDA) for generative classifiers (Lee et al., 2018). For task distributions, the variances $\boldsymbol{\Sigma}_1^{\mathsf{t}}, ..., \boldsymbol{\Sigma}_r^{\mathsf{t}}$ can be automatically inferred by our algorithm in Sec. 3.2.

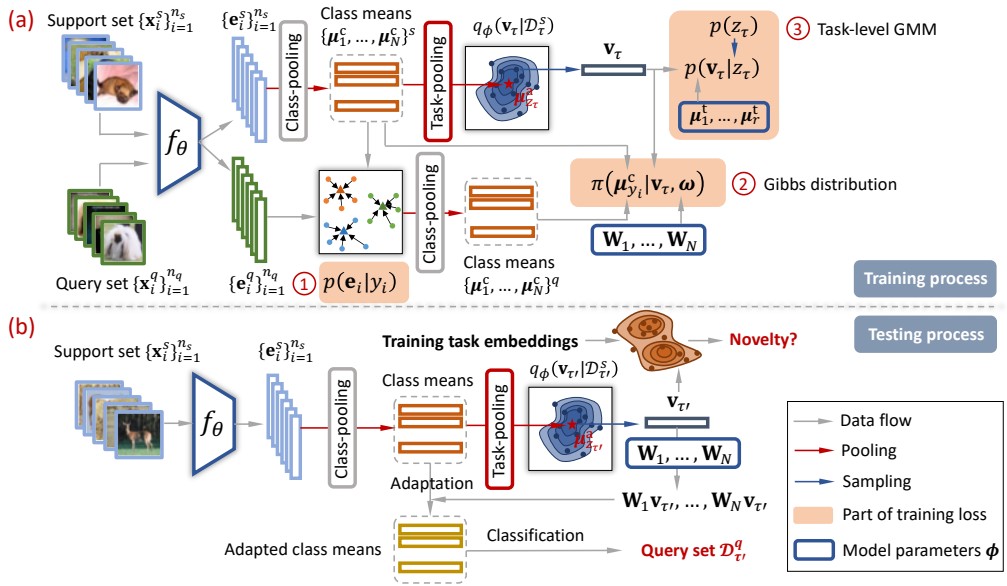

Figure 1: An illustration of HTGM on its (a) the training process, and (b) the testing process. In (a), ①②③ are the three parts of the training loss in Eq. (3). In (b), the training task embeddings contain the embeddings of all training tasks, *i.e.* the outputs of the task-pooling in (a).

Finally, substituting $p_{\boldsymbol{\theta}}(\mathbf{e}_i|y_i) = \mathcal{N}(\mathbf{e}_i|\boldsymbol{\mu}_{y_i}^{\mathrm{c}}, \sigma^2\mathbf{I})$, $p_{\boldsymbol{\omega}}(y_i|\mathbf{v}_\tau) = \pi(\boldsymbol{\mu}_{y_i}^{\mathrm{c}}|\mathbf{v}_\tau, \boldsymbol{\omega})$ $(y_i = k)$, $p(\mathbf{v}_\tau|z_\tau) = \mathcal{N}(\mathbf{v}_\tau|\boldsymbol{\mu}_{z_\tau}^{\mathrm{t}}, \boldsymbol{\Sigma}_{z_\tau}^{\mathrm{t}})$ and $p(z_\tau) = \mathrm{Uniform}(r)$ in Eq. (1), whose probabilities are specified and parameterized, we get our HTGM induced loss $\ell_{\mathrm{HTGM}}(\mathcal{D}_\tau; \boldsymbol{\theta}, \boldsymbol{\omega})$. The class means $\boldsymbol{\mu}_{y_i}^{\mathrm{c}}$, task means $\boldsymbol{\mu}_{z_\tau}^{\mathrm{t}}$ and variances $\boldsymbol{\Sigma}_{z_\tau}^{\mathrm{t}}$ are inferred in the E-step of our EM algorithm (details are in Sec. 3.2 and A.4).

## 3.2 MODEL OPTIMIZATION

It is hard to directly optimize $\ell_{\mathrm{HTGM}}(\mathcal{D}_\tau; \boldsymbol{\theta}, \boldsymbol{\omega})$, because the exact posterior inference is intractable (due to the integration over $\mathbf{v}_\tau$). To solve it, we resort to variational methods, and introduce an approximated posterior $q_{\boldsymbol{\phi}}(\mathbf{v}_\tau|\mathcal{D}_\tau^{\mathrm{s}})$, which is defined by an inference network $\phi$, and implies we want to infer $\mathbf{v}_\tau$ from its observed support set $\mathcal{D}_\tau^{\mathrm{s}}$. The query set $\mathcal{D}_\tau^{\mathrm{q}}$ is not included because it is unavailable during model testing. Then we propose to maximize a lower-bound of Eq. (1), which is derived as (the details are in Appendix A.1)

$$\ell_{\mathrm{HTGM}}(\mathcal{D}_\tau; \boldsymbol{\theta}, \boldsymbol{\omega}) \geq \ell_{\mathrm{HTGM\text{-}ELBO}}(\mathcal{D}_\tau; \boldsymbol{\theta}, \boldsymbol{\omega}) = \frac{1}{n}\sum_{i=1}^{n}\log p_{\boldsymbol{\theta}, \boldsymbol{\omega}}(\mathbf{e}_i|y_i)$$

$$+ \frac{1}{n}\sum_{i=1}^{n}\mathbb{E}_{\mathbf{v}_\tau \sim q_{\boldsymbol{\phi}}(\mathbf{v}_\tau|\mathcal{D}_\tau^{\mathrm{s}})}\Big[\log p_{\boldsymbol{\omega}}(y_i|\mathbf{v}_\tau) + \log\big(\sum_{z_\tau=1}^{r} p(\mathbf{v}_\tau|z_\tau)p(z_\tau)\big)\Big] + H\big(q_{\boldsymbol{\phi}}(\mathbf{v}_\tau|\mathcal{D}_\tau^{\mathrm{s}})\big) \tag{3}$$

where $H(q_{\boldsymbol{\phi}}(\mathbf{v}_\tau|\mathcal{D}_\tau^{\mathrm{s}})) = -\int_{\mathbf{v}_\tau} q_{\boldsymbol{\phi}}(\mathbf{v}_\tau|\mathcal{D}_\tau^{\mathrm{s}})\log q_{\boldsymbol{\phi}}(\mathbf{v}_\tau|\mathcal{D}_\tau^{\mathrm{s}})d\mathbf{v}_\tau$ is the entropy function. Similar to VAE (Kingma & Welling, 2013), Eq. (3) estimates the expectation (in the second term) by sampling $\mathbf{v}_\tau$ from $q_{\boldsymbol{\phi}}(\mathbf{v}_\tau|\mathcal{D}_\tau^{\mathrm{s}})$, instead of the integration in Eq. (1), hence facilitates computation. Next, we elaborate on the inference network, the challenges of maximizing Eq. (3), and our workarounds.

**Inference Network.** Similar to VAE, $q_{\boldsymbol{\phi}}(\mathbf{v}_\tau|\mathcal{D}_\tau^{\mathrm{s}})$ is defined as a Gaussian distribution $\mathcal{N}(\boldsymbol{\mu}_{z_\tau}^{\mathrm{a}}, \bar{\sigma}^2\mathbf{I})$, where $\boldsymbol{\mu}_{z_\tau}^{\mathrm{a}}$ is the output of the inference network, which approximates $\boldsymbol{\mu}_{z_\tau}^{\mathrm{t}}$ in Step 2 of the generative process, and $\bar{\sigma}$ is a hyperparameter for the corresponding variance. As illustrated by Fig. 1(a), the inference network is built upon the base model $f_{\boldsymbol{\theta}}(\cdot)$ with two non-parametric aggregation (*i.e.*, mean pooling) functions, thus $\phi = \boldsymbol{\theta}$. The first function aggregates class-wise embeddings to prototypes $\boldsymbol{\mu}_{y_i}^{\mathrm{c}}$'s, similar to prototypical networks (Snell et al., 2017). Differently, the second aggregates all prototypes to $\boldsymbol{\mu}_{z_\tau}^{\mathrm{a}}$. During model training, we use the reparameterization trick (Kingma & Welling, 2013) to sample $\mathbf{v}_\tau$ from $\mathcal{N}(\boldsymbol{\mu}_{z_\tau}^{\mathrm{a}}, \bar{\sigma}^2\mathbf{I})$. It is noteworthy that $H(q_{\boldsymbol{\phi}}(\mathbf{v}_\tau|\mathcal{D}_\tau^{\mathrm{s}}))$ in Eq. (3) becomes a constant now because $\bar{\sigma}^2$ is a constant.

**Challenge 1: Trivial Solution.** In Eq. (3), since the first term $\log p_{\boldsymbol{\theta},\boldsymbol{\omega}}(\mathbf{e}_i|y_i) = -\frac{1}{2\sigma^{2d}}\|\mathbf{e}_i - \boldsymbol{\mu}_{y_i}^{\mathsf{c}}\|_2^2$ (constants are ignored) only penalizes the distance between a sample $\mathbf{e}_i$ and its own class mean $\boldsymbol{\mu}_{y_i}^{\mathsf{c}}$ (*i.e.*, intra-class distances) without considering inter-class relationships, different class means $\boldsymbol{\mu}_1^{\mathsf{c}}$, ..., $\boldsymbol{\mu}_N^{\mathsf{c}}$ in task $\tau$ could collide, drawing all sample embeddings to the same spot. To avoid such a trivial solution and improve the stability of optimization, we apply negative sampling (Mikolov et al., 2013)

$$\ell_{\text{neg}}(\mathcal{D}_\tau; y_i, \boldsymbol{\theta}, \boldsymbol{\omega}) = -\log \mathbb{E}_{\mathbf{e}_j \sim \mathcal{D}_\tau}\left[\exp\left(-\frac{1}{2\sigma^{2d}}\|\mathbf{e}_j - \boldsymbol{\mu}_{y_i}^{\mathsf{c}}\|_2^2\right)\right] \quad (4)$$

where $\mathbf{e}_j$ is a negative sample embedding from any class in the support set, and $\boldsymbol{\mu}_{y_i}^{\mathsf{c}}$ is the mean of the positive class. In practice, we found it is beneficial to integrate $\ell_{\text{neg}}$ with our likelihood $\ell_{\text{HTGM}}$ in Eq. (1) during training, *i.e.* $\ell_{\text{HTGM}} + \frac{1}{n}\sum_{i=1}^{n}\ell_{\text{neg}}$. Correspondingly, from Eq. (3) we have

$$\ell(\mathcal{D}_\tau; \boldsymbol{\theta}, \boldsymbol{\omega}) = \ell_{\text{HTGM-ELBO}}(\mathcal{D}_\tau; \boldsymbol{\theta}, \boldsymbol{\omega}) + \frac{1}{n}\sum_{i=1}^{n}\ell_{\text{neg}}(\mathcal{D}_\tau; y_i, \boldsymbol{\theta}, \boldsymbol{\omega}) \quad (5)$$

which does not only serve as a robust training loss, but also helps solve the next challenge.

**Challenge 2: The Partition Function in Eq. (2).** The second term $p_{\boldsymbol{\omega}}(y_i|\mathbf{v}_\tau)$ in Eq. (3) involves computing the partition function in Eq. (2) (*i.e.*, the denominator), which is intractable because of the integration over all possible $\boldsymbol{\mu}_k^{\mathsf{c}}$'s. To solve it, we propose an upper bound of the partition function $\int_{\boldsymbol{\mu}_k} \exp\left[-E_{\boldsymbol{\omega}}(\boldsymbol{\mu}_k^{\mathsf{c}}; \mathbf{v}_\tau)\right]d\boldsymbol{\mu}_k^{\mathsf{c}} \leq N\sqrt{2^{d-1}\pi^d}$ (the derivation is in Appendix A.2), which is a constant with a specific $N$. By replacing the partition function in Eq. (2) with $N\sqrt{2^{d-1}\pi^d}$, we got a lower bound of $p_{\boldsymbol{\omega}}(y_i|\mathbf{v}_\tau)$, which in turn relaxes the lower bound in Eq. (3). The following theorem (the proof is in Appendix A.3) states the tightness of the relaxed bound is controllable.

**Theorem 1.** *Among the $N$ global maximums $\mathbf{W}_1\mathbf{v}_\tau, ..., \mathbf{W}_N\mathbf{v}_\tau$ of Eq. (2), let $\mathbf{W}_h\mathbf{v}_\tau$ and $\mathbf{W}_l\mathbf{v}_\tau$ ($1 \leq h, l \leq N$) be the pair with the smallest Euclidean distance $D(\mathbf{W}_h\mathbf{v}_\tau, \mathbf{W}_l\mathbf{v}_\tau)$, we have*

$$\lim_{D(\mathbf{W}_h\mathbf{v}_\tau, \mathbf{W}_l\mathbf{v}_\tau) \to \infty} \int_{\boldsymbol{\mu}_k} \exp\left[-E_{\boldsymbol{\omega}}(\boldsymbol{\mu}_k^{\mathsf{c}}; \mathbf{v}_\tau)\right]d\boldsymbol{\mu}_k^{\mathsf{c}} = N\sqrt{2^{d-1}\pi^d} \quad (6)$$

This theorem indicates the partition function approximates $N\sqrt{2^{d-1}\pi^d}$ when all pairs of the global maximums are far apart. It is noteworthy that during training (*i.e.*, maximizing the likelihood) we fit $\mathbf{W}_1\mathbf{v}_\tau, ..., \mathbf{W}_N\mathbf{v}_\tau$ to the different class prototypes $\boldsymbol{\mu}_1^{\mathsf{c}}, ..., \boldsymbol{\mu}_N^{\mathsf{c}}$ in $N$-way tasks. Because $\ell_{\text{neg}}$ in Eq. (4) tends to maximize the distances between different prototypes through the negative samples, maximizing the joint loss $\ell$ in Eq. (5) tends to separate $\mathbf{W}_1\mathbf{v}_\tau, ..., \mathbf{W}_N\mathbf{v}_\tau$, thus tighten the relaxed bound after using $N\sqrt{2^{d-1}\pi^d}$ according to Theorem 1. This is another benefit of negative sampling.

**Optimization via Expectation-Maximization.** In the third term of $\ell_{\text{HTGM-ELBO}}$ in Eq. (3), we need to estimate the mixture distribution $p(z_\tau)$. Similar to optimizing Gaussian mixture models, we alternately infer $p(z_\tau)$ and solve the model parameters $\{\boldsymbol{\theta}, \boldsymbol{\omega}\}$ through an Expectation-Maximization algorithm. In E-step, we infer $p(z_\tau)$ when fixing model parameters. In M-step, when fixing $p(z_\tau)$, $\{\boldsymbol{\theta}, \boldsymbol{\omega}\}$ can be efficiently solved by optimizing Eq. (5) with stochastic gradient descent (SGD). The formula to infer $p(z_\tau)$ and the detailed training algorithm of HTGM can be found in Appendix A.4.

### 3.3 MODEL ADAPTATION

Fig. 1(b) illustrates the adaptation process of HTGM. Given a new $N$-way task $\tau'$ from the meta-test set $\mathcal{D}^{\text{te}}$, its support set $\mathcal{D}_{\tau'}^{\mathsf{s}}$ is fed to the inference network to generate (1) class prototypes $\boldsymbol{\mu}_1^{\mathsf{c}}$, ..., $\boldsymbol{\mu}_N^{\mathsf{c}}$ (similar to prototypical networks), and (2) distribution $q_\phi(\mathbf{v}_{\tau'}|\mathcal{D}_{\tau'}^{\mathsf{s}})$, from which we draw the average task embedding $\mathbf{v}_{\tau'} = \boldsymbol{\mu}_{z_{\tau'}}^{\mathsf{a}}$. Recall that the inference network is the base model $f_{\boldsymbol{\theta}}(\cdot)$ with class-pooling and task-pooling layers, as illustrated in Fig. 1(b), and $\phi = \boldsymbol{\theta}$. Then $\mathbf{v}_{\tau'}$ is projected to $\mathbf{W}_1\mathbf{v}_{\tau'}, ..., \mathbf{W}_N\mathbf{v}_{\tau'}$ which represent the $N$ optimal choices of class prototypes for task $\tau'$ as learned by the Gibbs distribution in Eq. (2) from the training tasks. They are used to adapt $\boldsymbol{\mu}_1^{\mathsf{c}}, ..., \boldsymbol{\mu}_N^{\mathsf{c}}$ so that the adapted prototypes are drawn towards the closest classes from the mixture component that task $\tau'$ belongs to. The adaptation is performed by selecting the closest optimum for each prototype, *i.e.*, $\bar{\boldsymbol{\mu}}_j^{\mathsf{c}} = \alpha\boldsymbol{\mu}_j^{\mathsf{c}} + (1-\alpha)\mathbf{W}_{l^*}\mathbf{v}_{\tau'}$ where $l^* = \arg\min_{1 \leq l \leq N} D(\boldsymbol{\mu}_j^{\mathsf{c}}, \mathbf{W}_l\mathbf{v}_{\tau'})$ using Euclidean distance $D(\cdot, \cdot)$ and $\alpha$ is a hyperparameter. Finally, we (1) assess if $\tau'$ is a novelty by computing the likelihood of $\mathbf{v}_{\tau'}$ in a pre-fitted GMM on the embeddings $\mathbf{v}_\tau$'s of the training tasks in $\mathcal{D}^{\text{tr}}$, and (2) perform classification on each sample $\mathbf{x}_i'$ in the query set $\mathcal{D}_{\tau'}^{\mathsf{q}}$ using the adapted prototypes by $p(y_i' = j'|\mathbf{x}_i') = \frac{\exp\left(-D(f_{\boldsymbol{\theta}}(\mathbf{x}_i'), \bar{\boldsymbol{\mu}}_{j'}^{\mathsf{c}})\right)}{\sum_{j=1}^{N}\exp\left(-D(f_{\boldsymbol{\theta}}(\mathbf{x}_i'), \bar{\boldsymbol{\mu}}_j^{\mathsf{c}})\right)}$.

| Setting | Model | Bird | Texture | Aircraft | Fungi | Average |
|---|---|---|---|---|---|---|
| 5-way 1-shot | TAML | 55.77±1.43 | 31.78±1.30 | 48.56±1.37 | 41.00±1.50 | 44.28 |
| | MAML | 53.94±1.45 | 31.66±1.31 | 51.37±1.38 | 42.12±1.36 | 44.77 |
| | Meta-SGD | 55.58±1.43 | 32.38±1.32 | 52.99±1.36 | 41.74±1.34 | 45.67 |
| | MUMOMAML | 56.82±1.49 | 33.81±1.36 | 53.14±1.39 | 42.22±1.40 | 46.50 |
| | HSML | 60.98±1.50 | 35.01±1.36 | 57.38±1.40 | 44.02±1.39 | 49.35 |
| | ARML | 62.33±1.47 | 35.65±1.40 | 58.56±1.41 | 44.82±1.38 | 50.34 |
| | ProtoNet | 61.54±1.27 | 38.84±1.42 | 73.42±1.23 | 46.52±1.42 | 55.08 |
| | MetaOptNet | 62.80±1.29 | 44.30±1.45 | 68.64±1.29 | 47.04±1.38 | 55.70 |
| | ProtoNet-Aug | 65.04±1.29 | 44.68±1.43 | 70.44±1.32 | 49.30±1.40 | 57.37 |
| | NCA | 62.58±1.25 | 40.98±1.44 | 68.70±1.26 | 46.36±1.34 | 54.66 |
| | FEATS | 62.60±1.31 | 44.12±1.49 | 68.86±1.28 | 47.92±1.34 | 55.88 |
| | HTGM (ours) | **70.12±1.28** | **47.76±1.49** | **75.52±1.24** | **52.06±1.41** | **61.37** |
| 5-way 5-shot | TAML | 69.50±0.75 | 45.11±0.69 | 65.92±0.74 | 50.99±0.87 | 57.88 |
| | MAML | 68.52±0.79 | 44.56±0.68 | 66.18±0.71 | 51.85±0.85 | 57.78 |
| | Meta-SGD | 67.87±0.74 | 45.49±0.68 | 66.84±0.70 | 52.51±0.81 | 58.18 |
| | MUMOMAML | 70.49±0.76 | 45.89±0.69 | 67.31±0.68 | 53.96±0.82 | 59.41 |
| | HSML | 71.68±0.73 | 48.08±0.69 | 73.49±0.68 | 56.32±0.80 | 62.39 |
| | ARML | 73.34±0.70 | 49.67±0.67 | 74.88±0.64 | 57.55±0.82 | 63.86 |
| | ProtoNet | 78.88±0.72 | 57.93±0.75 | 86.42±0.57 | 62.52±0.79 | 71.44 |
| | MetaOptNet | 81.66±0.71 | **61.97±0.78** | 84.03±0.56 | 63.80±0.81 | 72.87 |
| | ProtoNet-Aug | 80.62±0.71 | 58.30±0.77 | 87.05±0.53 | 63.62±0.81 | 72.39 |
| | NCA | 79.16±0.75 | 58.69±0.76 | 85.27±0.53 | 61.68±0.80 | 71.20 |
| | FEATS | 78.37±0.72 | 57.02±0.73 | 85.55±0.54 | 61.56±0.80 | 70.63 |
| | HTGM (ours) | **82.27±0.74** | 60.67±0.78 | **88.48±0.52** | **65.70±0.79** | **74.28** |

Table 1: Results (accuracy±95% confidence) of the compared methods on Plain-Multi dataset.

## 4 EXPERIMENTS

In this section, we evaluate HTGM's effectiveness on few-shot classification and novel task detection on benchmark datasets, and compare it with SOTA methods.

**Datasets.** The first is the *Plain-Multi* benchmark proposed in (Yao et al., 2019a). It includes four fine-grained image classification datasets, *i.e.*, CUB-200-2011 (Bird), Describable Textures Dataset (Texture), FGVC of Aircraft (Aircraft), and FGVCx-Fungi (Fungi). In each episode, a task samples classes from one of the four datasets, so that different tasks are from a mixture of the four domains. The second is the *Art-Multi* benchmark from (Yao et al., 2019b), whose distribution is more complex than Plain-Multi. Similar to (Jerfel et al., 2019), each image in Plain-Multi was applied with two filters, *i.e.*, *blur* filter and *pencil* filter, respectively, to simulate a changing distribution of few-shot tasks. Afterward, together with the original four datasets, a total of 12 datasets comprise Art-Multi, and each task is sampled from one of them. Both benchmarks were divided into the meta-training, meta-validation, and meta-test sets by following their corresponding papers.

**Baselines.** We compare HTGM with the most relevant SOTA methods on meta-learning, including (1) optimization-based methods: MAML (Finn et al., 2017) and Meta-SGD (Li et al., 2017) learn globally shared initialization among tasks. MUMOMAML (Vuorio et al., 2018) is a task-specific method. TAML (Lee et al., 2019a) handles imbalanced tasks. HSML (Yao et al., 2019a) and ARML (Yao et al., 2019b) learn locally shared initial parameters in clusters of tasks and neighborhoods of a meta-graph of tasks, respectively; and (2) Metric-based methods: ProtoNet (Snell et al., 2017) learns prototypes with distance-based classifier. MetaOptNet (Lee et al., 2019c) uses an SVM classifier with kernel metrics. ProtoNet-Aug (Su et al., 2020), FEATS (Ye et al., 2020) and NCA (Laenen & Bertinetto, 2021) were built upon ProtoNet by augmenting images (*e.g.*, rotation, jigsaw), adding prototype aggregator (*e.g.*, Transformer), and using contrastive training loss, (instead of prototype-based loss), respectively. The detailed setup of these methods is deferred to Appendix C.1.

**Implementation.** Following (Tian et al., 2020), optimization-based baselines use the standard four-block convolutional layers as the base learner, and metric-based methods use ResNet-12 as the base learner. The output dimension of these networks is 640 (MetaOptNet uses 16000 as in its paper). In our experiments, we observed the optimization-based methods have out-of-memory issues when using ResNet-12, indicating their limitation in using large backbone networks. To test them

| Setting | Model | Original | Blur | Pencil | Average |
|---|---|---|---|---|---|
| | TAML | 42.22±1.39 | 40.02±1.41 | 35.11±1.34 | 39.11 |
| | MAML | 42.70±1.35 | 40.53±1.38 | 36.71±1.37 | 39.98 |
| | Meta-SGD | 44.21±1.38 | 42.36±1.39 | 37.21±1.39 | 41.26 |
| | MUMOMAML | 45.63±1.39 | 41.59±1.38 | 39.24±1.36 | 42.15 |
| | HSML | 47.92±1.34 | 44.43±1.34 | 41.44±1.34 | 44.60 |
| 5-way,1-shot | ARML | 45.68±1.34 | 42.62±1.34 | 39.78±1.34 | 42.69 |
| | ProtoNet | 55.23±1.31 | 51.70±1.42 | 49.22±1.44 | 52.05 |
| | MetaOptNet | 56.10±1.35 | 52.33±1.43 | 49.08±1.45 | 52.50 |
| | ProtoNet-Aug | 57.63±1.34 | 55.00±1.40 | 49.73±1.53 | 54.12 |
| | NCA | 56.12±1.35 | 50.80±1.49 | 47.99±1.45 | 51.64 |
| | FEATS | 54.33±1.33 | 50.90±1.48 | 47.96±1.48 | 51.07 |
| | HTGM (ours) | **61.18±1.34** | **58.80±1.42** | **53.23±1.48** | **57.74** |
| | TAML | 58.54±0.73 | 55.23±0.75 | 49.23±0.75 | 54.33 |
| | MAML | 58.30±0.74 | 55.71±0.74 | 49.59±0.73 | 54.50 |
| | Meta-SGD | 57.82±0.72 | 55.54±0.73 | 50.24±0.72 | 54.53 |
| | MUMOMAML | 58.60±0.75 | 56.29±0.72 | 51.15±0.73 | 55.35 |
| | HSML | 60.63±0.73 | 57.91±0.72 | 53.93±0.72 | 57.49 |
| 5-way,1-shot | ARML | 61.78±0.74 | 58.73±0.75 | 55.27±0.73 | 58.59 |
| | ProtoNet | 71.34±0.73 | 67.28±0.75 | 64.32±0.76 | 67.65 |
| | MetaOptNet | 72.33±0.72 | 68.90±0.78 | 63.89±0.71 | 68.37 |
| | ProtoNet-Aug | 72.87±0.71 | 70.50±0.72 | 63.98±0.73 | 68.78 |
| | NCA | 72.44±0.72 | 67.33±0.71 | 62.98±0.78 | 67.58 |
| | FEATS | 71.99±0.71 | 67.54±0.72 | 63.09±0.76 | 67.54 |
| | HTGM (ours) | **74.67±0.70** | **71.24±0.73** | **65.22±0.77** | **70.37** |

Table 2: Results (accuracy±95% confidence) of the compared methods on Art-Multi dataset.

on ResNet-12, we followed the ANIL method (Raghu et al., 2020) by pre-training ResNet-12 via ProtoNet, freezing the encoder, and fine-tuning the last fully-connected layer. In this case, HSML and ARML cannot work properly as they require joint training of the encoder and other layers. The details are in Appendix D.5. For training, Adam optimizer was used. Each batch contains 4 tasks. Each model was trained with 20000 episodes. The learning rate of metric-based methods is $1e^{-3}$. The learning rates for inner- and outer-loops for optimization-based methods are $1e^{-3}$ and $1e^{-4}$. The weight decay was $1e^{-4}$. For HTGM, we set $\sigma = 1.0, \bar{\sigma} = 0.1, \alpha = 0.5$ (0.9) for 1-shot (5-shot) tasks. The number of mixture components $r$ varies *w.r.t.* different datasets, and was grid-searched within $[2, 4, 8, 16, 32]$. All hyperparameters were set according to the meta-validation sets.

### 4.1 EXPERIMENTAL RESULTS

**Few-shot classification.** Following (Tian et al., 2020), we report the mean accuracy and 95% confidence interval of 1000 random tasks with 5-way 1-shot/5-shot, 5/25-query tests. Following (Yao et al., 2019b), we report the accuracy of each domain (Bird, Texture, Aircraft and Fungi) and the overall average accuracy for Plain-Multi, and report the accuracy of each image filtering strategy and the overall average accuracy for Art-Multi.

Table 1 and 2 summarize the results. From the tables, we have several observations. First, metric-based methods generally outperform optimization-based methods. This is because of the efficiency of metric-based methods, enabling them to fit a larger backbone network, which is consistent with the results in (Tian et al., 2020). Built upon the metric-based method, HTGM only introduces a few distribution-related parameters and thus has the flexibility to scale with the encoder size. Second, baselines designed for dealing with mixture distributions of tasks, *i.e.*, HSML and ARML, outperform their counterparts without such design, demonstrating the importance to consider mixture task distribution in practice. Finally, HTGM outperforms the SOTA baselines in most cases by large margins, suggesting its effectiveness in modeling the generative process of task instances.

**Novel task detection.** We also evaluate HTGM on the task of detecting novel $N$-way-$K$-shot tasks ($N = 5$, $K = 1$) that are drawn out of the training task distributions. To this end, we train each comapred model in the Original domain in Art-Multi dataset, and test the model on tasks drawn from either Original domain (*i.e.*, known tasks), or {Blur, Pencil} domains (*i.e.*, novel tasks), and evaluate if the model can tell whether a testing task is known or novel.

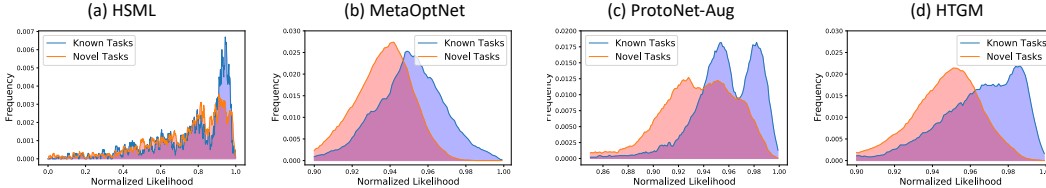

Figure 2: The frequency of tasks *w.r.t.* the normalized likelihood for (a) HSML (b) MetaOptNet (c) ProtoNet-Aug (d) HTGM. The x-axis ranges vary as only 95% tasks with top scores were preserved.

For comparison, since none of the baselines detects novel tasks, we adapt them as follows. For metric-based methods, since they use a fixed encoder for all training/testing tasks, we averaged the sample embeddings in each task to represent the task. Then a separate GMM model was built upon the training task embeddings, and its likelihood was adapted to score the novelty of testing tasks (some details of setup are in Appendix C.2. However, optimization-based models perform gradient descent on the support set of each task, leading to varying encoders per task. As such, sample embeddings of different tasks are not comparable, and we cannot obtain task embeddings in the same way as before. Among them, only HSML has an augmented task-level encoder for task embedding, allowing us to include it for comparison. For a fair comparison, our HTGM also trains a GMM on its task embeddings for detecting novel tasks. Moreover, two HTGM variants were included for ablation analysis to understand some design choices: (1) HTGM-Gaussian replaces the Gibbs distribution in Eq. (2) with a Gaussian distribution; (2) HTGM w/o GMM removes the task-level GM, *i.e.*, the third term in Eq. (3). The classification results of the ablation variants are in Appendix D.4. Following (Cheng & Vasconcelos, 2021; Vaze et al., 2022; Sharma et al., 2021), we report Area Under ROC (AUROC), Average Precision (AP), and Max-F1 for performance evaluation.

Table 3 summarize the results, from which we observe HTGM outperforms all baselines over all evaluation metrics, indicating the superior quality of task embeddings learned by our model. The embeddings follow the specified mixture distribution of tasks $p(\mathbf{v}_\tau)$ as described in Sec. 3.1, which fits the mixture data well hence allowing to detect novel tasks that are close to the boundary. Since the baselines learn embeddings without explicit constraint, they even don't fit the post-hoc GMM very well. Moreover, HTGM outperforms HTGM w/o GMM, which is even worse than some other baselines. This further validates the necessity to introduce the regularization of task-level mixture distribution $p(\mathbf{v}_\tau)$. Also, the drops of

| Model | AUROC | AP | Max-F1 |
|---|---|---|---|
| HSML | 55.96 | 37.94 | 50.17 |
| ProtoNet | 65.17 | 41.51 | 56.07 |
| MetaOptNet | 72.71 | 63.77 | 58.33 |
| NCA | 66.28 | 51.45 | 52.74 |
| ProtoNet-Aug | 72.67 | 57.93 | 59.07 |
| FEATS | 59.35 | 42.57 | 49.31 |
| HTGM w/o GMM | 70.24 | 62.45 | 57.75 |
| HTGM-Gaussian | 74.06 | 66.18 | **60.62** |
| HTGM | **75.66** | **68.03** | 60.51 |

Table 3: Comparison between HTGM and its variants and the applicable baselines on novel task detection.

AUROC and AP of HTGM-Gaussian demonstrate the importance of our unique design of the Gibbs distribution for the task-conditional distribution in Eq. (2). Similar to (Vaze et al., 2022), in Fig. 2, we visualized the normalized likelihood histogram of known and novel tasks for HSML, MetaOptNet (the best baseline), ProtoNet-Aug (the near-best baseline), and HTGM. The figures indicate the likelihoods (*i.e.*, novelty scores) of HTGM are more distinguishable for known and novel tasks than the baselines. We also analyzed the hyperparameters of HTGM, which are in D.1, D.2, D.3.

## 5 CONCLUSION

In this paper, we propose a novel Hierarchical Gaussian Mixture based Task Generative Model (HTGM). HTGM models the generative process of task instances, and performs maximum likelihood estimation to learn task embeddings, which can help adjust prototypes acquired by the feature extractor and thus achieve better performance. Moreover, by explicitly modeling the mixture distribution of tasks in the embedding space, HTGM can effectively detect the tasks that are drawn from distributions unseen in the meta-training stage. The extensive experimental results indicate the advantage of the proposed method on both few-shot classification and novel task detection.

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

# A APPENDIX FOR DETAILS OF DERIVING HTGM

## A.1 THE LOWER-BOUND OF THE LIKELIHOOD FUNCTION

In this section, we provide the details of the lower-bound in Eq. (3). By introducing the approximated posterior $q_\phi(\mathbf{v}_\tau|\mathcal{D}_\tau^\mathsf{s})$, the likelihood in Eq. (1) becomes (the superscript $*$ is neglected for clarity)

$$
\begin{aligned}
\ell(\mathcal{D}_\tau, \boldsymbol{\theta}) &= \frac{1}{n}\sum_{i=1}^n \log p_{\boldsymbol{\theta}}(\mathbf{e}_i|y_i) + \frac{1}{n}\sum_{i=1}^n \log \Big( \int_{\mathbf{v}_\tau} p(y_i|\mathbf{v}_\tau)p(\mathbf{v}_\tau)d\mathbf{v}_\tau \Big) \\
&= \frac{1}{n}\sum_{i=1}^n \log p_{\boldsymbol{\theta}}(\mathbf{e}_i|y_i) + \frac{1}{n}\sum_{i=1}^n \log \Big( \int_{\mathbf{v}_\tau} p(y_i|\mathbf{v}_\tau)p(\mathbf{v}_\tau)\frac{q_\phi(\mathbf{v}_\tau|\mathcal{D}_\tau^\mathsf{s})}{q_\phi(\mathbf{v}_\tau|\mathcal{D}_\tau^\mathsf{s})}d\mathbf{v}_\tau \Big) \\
&= \frac{1}{n}\sum_{i=1}^n \log p_{\boldsymbol{\theta}}(\mathbf{e}_i|y_i) + \frac{1}{n}\sum_{i=1}^n \log \Big( \int_{\mathbf{v}_\tau} q_\phi(\mathbf{v}_\tau|\mathcal{D}_\tau^\mathsf{s})\frac{p(y_i|\mathbf{v}_\tau)p(\mathbf{v}_\tau)}{q_\phi(\mathbf{v}_\tau|\mathcal{D}_\tau^\mathsf{s})}d\mathbf{v}_\tau \Big) \\
&\geq \frac{1}{n}\sum_{i=1}^n \log p_{\boldsymbol{\theta}}(\mathbf{e}_i|y_i) + \frac{1}{n}\sum_{i=1}^n \int_{\mathbf{v}_\tau} q_\phi(\mathbf{v}_\tau|\mathcal{D}_\tau^\mathsf{s})\Big[ \log p(y_i|\mathbf{v}_\tau) + \log p(\mathbf{v}_\tau) - \log q_\phi(\mathbf{v}_\tau|\mathcal{D}_\tau^\mathsf{s}) \Big]d\mathbf{v}_\tau \\
&= \frac{1}{n}\sum_{i=1}^n \log p_{\boldsymbol{\theta}}(\mathbf{e}_i|y_i) + \frac{1}{n}\sum_{i=1}^n \int_{\mathbf{v}_\tau} q_\phi(\mathbf{v}_\tau|\mathcal{D}_\tau^\mathsf{s})\Big[ \log p(y_i|\mathbf{v}_\tau) + \log p(\mathbf{v}_\tau) \Big]d\mathbf{v}_\tau - \int_{\mathbf{v}_\tau} q_\phi(\mathbf{v}_\tau|\mathcal{D}_\tau^\mathsf{s}) \log q_\phi(\mathbf{v}_\tau|\mathcal{D}_\tau^\mathsf{s})d\mathbf{v}_\tau \\
&= \frac{1}{n}\sum_{i=1}^n \log p_{\boldsymbol{\theta}}(\mathbf{e}_i|y_i) + \frac{1}{n}\sum_{i=1}^n \mathbb{E}_{\mathbf{v}_\tau \sim q_\phi(\mathbf{v}_\tau|\mathcal{D}_\tau^\mathsf{s})}\Big[ \log p(y_i|\mathbf{v}_\tau) + \log p(\mathbf{v}_\tau) \Big] + H(q_\phi(\mathbf{v}_\tau|\mathcal{D}_\tau^\mathsf{s})) \\
&= \frac{1}{n}\sum_{i=1}^n \log p_{\boldsymbol{\theta}}(\mathbf{e}_i|y_i) + \frac{1}{n}\sum_{i=1}^n \mathbb{E}_{\mathbf{v}_\tau \sim q_\phi(\mathbf{v}_\tau|\mathcal{D}_\tau^\mathsf{s})}\Big[ \log p(y_i|\mathbf{v}_\tau) + \log \Big( \sum_{z_\tau=1}^r p(\mathbf{v}_\tau|z_\tau)p(z_\tau) \Big) \Big] + H(q_\phi(\mathbf{v}_\tau|\mathcal{D}_\tau^\mathsf{s}))
\end{aligned}
$$
(7)

where the fourth step uses Jensen's inequality. This completes the derivation of Eq. (3).

## A.2 THE UPPER-BOUND OF THE PARTITION FUNCTION

In Sec. 3.2, we apply an upper bound on the partition function in Eq. (2) for solving the challenging 2. The derivation of the upper bound is as follows.

$$
\begin{aligned}
\int_{\boldsymbol{\mu}_{y_i}^\mathsf{c}} \exp\big[ -E_{\boldsymbol{\omega}}(\boldsymbol{\mu}_{y_i}^\mathsf{c}; \mathbf{v}_\tau) \big]d\boldsymbol{\mu}_{y_i}^\mathsf{c} &= \int_{\boldsymbol{\mu}_{y_i}^\mathsf{c}} \exp\big[ -\min(\{||\boldsymbol{\mu}_{y_i}^\mathsf{c} - \mathbf{W}_j\mathbf{v}_\tau||_2^2\}_{j=1}^N) \big]d\boldsymbol{\mu}_{y_i}^\mathsf{c} \\
&= \int_{\boldsymbol{\mu}_{y_i}^\mathsf{c}} \max\Big( \big\{ \exp\big[ -||\boldsymbol{\mu}_{y_i}^\mathsf{c} - \mathbf{W}_j\mathbf{v}_\tau||_2^2 \big] \big\}_{j=1}^N \Big)d\boldsymbol{\mu}_{y_i}^\mathsf{c} < \int_{\boldsymbol{\mu}_{y_i}^\mathsf{c}} \sum_{j=1}^N \exp\big[ -||\boldsymbol{\mu}_{y_i}^\mathsf{c} - \mathbf{W}_j\mathbf{v}_\tau||_2^2 \big]d\boldsymbol{\mu}_{y_i}^\mathsf{c} \\
&= \sum_{j=1}^N \int_{\boldsymbol{\mu}_{y_i}^\mathsf{c}} \exp\big[ -||\boldsymbol{\mu}_{y_i}^\mathsf{c} - \mathbf{W}_j\mathbf{v}_\tau||_2^2 \big]d\boldsymbol{\mu}_{y_i}^\mathsf{c} = \sqrt{2^{d-1}\pi^d}
\end{aligned}
$$
(8)

where the last equation is from the multidimensional Gaussian integral. This completes the derivation of the upper bound of the partition function.

## A.3 THE PROOF OF THEOREM 1

*Proof.* Let $B_j$ denote a ball in $\mathbb{R}^d$. Its center is at $\mathbf{W}_j\mathbf{v}_\tau$ and its radius is $D(\mathbf{W}_h\mathbf{v}_\tau, \mathbf{W}_l\mathbf{v}_\tau)/3$. Because $\mathbf{W}_h\mathbf{v}_\tau$ and $\mathbf{W}_l\mathbf{v}_\tau$ $(1 \leq h,l \leq N)$ is the pair with the smallest Euclidean distance $D(\mathbf{W}_h\mathbf{v}_\tau, \mathbf{W}_l\mathbf{v}_\tau)$, for any pair of balls $B_j$ and $B_m$ we have $B_j \cup B_m = \varnothing$.

In other words, there is no overlap between any pair of balls. Therefore, if we compute the integral over the joint of all balls, we have

$$
\int_{\boldsymbol{\mu}_k^\mathsf{c} \in \cup_{m=1}^N B_m} \exp\big[ -E_{\boldsymbol{\omega}}(\boldsymbol{\mu}_k^\mathsf{c}; \mathbf{v}_\tau) \big]d\boldsymbol{\mu}_k^\mathsf{c} = \sum_{m=1}^N \int_{\boldsymbol{\mu}_k^\mathsf{c} \in B_m} \exp\big[ -E_{\boldsymbol{\omega}}(\boldsymbol{\mu}_k^\mathsf{c}; \mathbf{v}_\tau) \big]d\boldsymbol{\mu}_k^\mathsf{c}
$$
(9)

Also, because there is no overlap between any pair of balls, for each point $\boldsymbol{\mu}_k^\mathsf{c} \in B_m$, we have

$$
-\min\Big( \{||\boldsymbol{\mu}_k^\mathsf{c} - \mathbf{W}_j\mathbf{v}_\tau||_2^2\}_{j=1}^N \Big) = -||\boldsymbol{\mu}_k^\mathsf{c} - \mathbf{W}_m\mathbf{v}_\tau||_2^2
$$
(10)

Therefore, we have the following derivation from Eq. (9).

$$\int_{\boldsymbol{\mu}_k^{\mathsf{c}} \in \bigcup_{m=1}^N B_m} \exp\big[-E_{\boldsymbol{\omega}}(\boldsymbol{\mu}_k^{\mathsf{c}}; \mathbf{v}_\tau)\big] d\boldsymbol{\mu}_k^{\mathsf{c}} = \sum_{m=1}^N \int_{\boldsymbol{\mu}_k^{\mathsf{c}} \in B_m} \exp\big[-E_{\boldsymbol{\omega}}(\boldsymbol{\mu}_k^{\mathsf{c}}; \mathbf{v}_\tau)\big] d\boldsymbol{\mu}_k^{\mathsf{c}}$$

$$= \sum_{m=1}^N \int_{\boldsymbol{\mu}_k^{\mathsf{c}} \in B_m} \exp\big[-||\boldsymbol{\mu}_k^{\mathsf{c}} - \mathbf{W}_m \mathbf{v}_\tau||_2^2\big] d\boldsymbol{\mu}_k^{\mathsf{c}} = N \int_{\boldsymbol{\mu}_k \in B_m} \exp\big[-||\boldsymbol{\mu}_k^{\mathsf{c}} - \mathbf{W}_m \mathbf{v}_\tau||_2^2\big] d\boldsymbol{\mu}_k^{\mathsf{c}} \tag{11}$$

Meanwhile, since $\bigcup_{m=1}^N B_m$ is a sub-area of the entire $\mathbb{R}^d$ space, we have

$$\int_{\boldsymbol{\mu}_k^{\mathsf{c}} \in \bigcup_{m=1}^N B_m} \exp\big[-E_{\boldsymbol{\omega}}(\boldsymbol{\mu}_k^{\mathsf{c}}; \mathbf{v}_\tau)\big] d\boldsymbol{\mu}_k^{\mathsf{c}} \leq \int_{\boldsymbol{\mu}_k^{\mathsf{c}}} \exp\big[-E_{\boldsymbol{\omega}}(\boldsymbol{\mu}_k^{\mathsf{c}}; \mathbf{v}_\tau)\big] d\boldsymbol{\mu}_k^{\mathsf{c}} \tag{12}$$

According to the multidimensional Gaussian integral, we have

$$\lim_{D(\mathbf{W}_h \mathbf{v}_\tau, \mathbf{W}_l \mathbf{v}_\tau) \to \infty} \int_{\boldsymbol{\mu}_k^{\mathsf{c}} \in B_m} \exp\big[-E_{\boldsymbol{\omega}}(\boldsymbol{\mu}_k^{\mathsf{c}}; \mathbf{v}_\tau)\big] d\boldsymbol{\mu}_k^{\mathsf{c}} = \sqrt{2^{d-1}\pi^d} \tag{13}$$

Therefore,

$$\lim_{D(\mathbf{W}_h \mathbf{v}_\tau, \mathbf{W}_l \mathbf{v}_\tau) \to \infty} \int_{\boldsymbol{\mu}_k^{\mathsf{c}}} \exp\big[-E_{\boldsymbol{\omega}}(\boldsymbol{\mu}_k^{\mathsf{c}}; \mathbf{v}_\tau)\big] d\boldsymbol{\mu}_k^{\mathsf{c}} \geq N\sqrt{2^{d-1}\pi^d} \tag{14}$$

Since $N\sqrt{2^{d-1}\pi^d}$ is its upper bound, based on the squeeze theorem, we have

$$\lim_{D(\mathbf{W}_h \mathbf{v}_\tau, \mathbf{W}_l \mathbf{v}_\tau) \to \infty} \int_{\boldsymbol{\mu}_k^{\mathsf{c}}} \exp\big[-E_{\boldsymbol{\omega}}(\boldsymbol{\mu}_k^{\mathsf{c}}; \mathbf{v}_\tau)\big] d\boldsymbol{\mu}_k^{\mathsf{c}} = N\sqrt{2^{d-1}\pi^d} \tag{15}$$

which completes the proof of Theorem 1. $\qquad\square$

### A.4 THE TRAINING ALGORITHM OF HTGM

The training algorithm of HTGM is summarized in Algorithm 1.

## B APPENDIX FOR FURTHER DISCUSSION

### B.1 DISCUSSION ABOUT THE RELATIONSHIP BETWEEN HTGM AND HGM MODEL

To the best of our knowledge, the Hierarchical Gaussian Mixture (HGM) model has appeared in the traditional works (Goldberger & Roweis, 2005; Olech & Paradowski, 2016; Athey et al., 2019) for hierarchical clustering by applying Gaussian Mixture model agglomeratively or divisively on the input samples. They are unsupervised methods that infer clusters of samples, but do not pre-train embedding models (or parameter initializations) that could be fine-tuned for the adaptation to new tasks in meta-learning. Therefore, these methods are remarkably different from meta-learning methods, and we think it is a non-trivial problem to adapt the concept of HGM to solve the meta-learning problem. To this end, we need to (1) identify the motivation; and (2) solve the new technical challenges. For (1), we found the hierarchical structure of mixture distributions naturally appears when we want to model the generative process of tasks from a mixture of distributions, where each task contains another mixture distribution of classes (as suggested by Eq. (1)). In other words, the motivating point of our method is more on meta-learning than HGM. However, we think drawing such a connection between meta-learning and HGM is a novel contribution. For (2), our method is different from traditional HGM in (a) its generative process of tasks (Sec. 3.1), which is a theoretical extension of the widely used empirical process of generating tasks in meta-learning; (b) its Gibbs-style task-conditional distribution (Eq. (2)) for fitting uniformly sampled classes; (c) the metric-based end-to-end meta-learning framework (Fig. 1) (note the traditional HGM is not for learning embeddings); (d) the non-trivial derivation of the optimization algorithm in Sect. 3.2 and Alg. 1; and (e) the novel model adaptation process in Sec. 3.3. Solving the technical challenges in the new generative model is another novel contribution of the proposed method.

---

**Algorithm 1:** Hierarchical Gaussian Mixture based Task Generative Model (HTGM)

---

**Input:** encoder $f_{\boldsymbol{\theta}}$, training dataset $\mathcal{D}^{\text{tr}}$, hyperparameters $r, \sigma, \bar{\sigma}$
**Output:** model parameters $\{\boldsymbol{\theta}, \boldsymbol{\omega}\}$

1 Pre-train the encoder $f_{\boldsymbol{\theta}}$ via ProtoNet with augmentations.

2 Pre-train the energy function in Eq. (2) by maximizing $\frac{1}{n}\sum_{i=1}^{n}\log p_{\boldsymbol{\theta},\boldsymbol{\omega}}(\mathbf{e}_i|y_i) + \log p_{\boldsymbol{\omega}}(y_i|\mathbf{v}_\tau)$

3 **for** $i \leftarrow 1$ *to MaxEpoch* **do**
    /* E-step */
4     $\mathcal{V} = \varnothing$
5     **for** $\{\mathcal{D}^s_\tau = \{(\mathbf{x}^s_i, y^s_i)\}_{i=1}^{ns}, \mathcal{D}^q_\tau = \{(\mathbf{x}^q_i, y^q_i)\}_{i=1}^{nq}\}$ *in Dataloader($\mathcal{D}^{tr}$)* **do**
          /* load a task episode */
6       $\{\mathbf{e}^s_i\}_{i=1}^{ns} = \{f_{\boldsymbol{\theta}}(\mathbf{x}^s_i)\}_{i=1}^{ns}$         // embeddings of the support set
7       $\boldsymbol{\mu}^a_{z_\tau} = $ Task-Pooling(Class-Pooling($\{(\mathbf{e}^s_i, y^s_i)\}_{i=1}^{ns}$))     // the mean of $q_{\boldsymbol{\phi}}(\mathbf{v}_\tau|\mathcal{D}^s_\tau)$
8       Sample a task embedding $\mathbf{v}_\tau$ from $q_{\boldsymbol{\phi}}(\mathbf{v}_\tau|\mathcal{D}^s_\tau) = \mathcal{N}(\boldsymbol{\mu}^a_{z_\tau}, \bar{\sigma}^2\mathbf{I})$
9       $\mathcal{V} = \mathcal{V} \cup \{\mathbf{v}_\tau\}$
10     **end**
11     $\{z_\tau\}_{\tau=1}^{|\mathcal{V}|}, \{\boldsymbol{\mu}^t_1, ..., \boldsymbol{\mu}^t_r, \boldsymbol{\Sigma}^t_1, ..., \boldsymbol{\Sigma}^t_r\} = $ GMM($\mathcal{V}$).     // fit a GMM to $\mathcal{V}$, where $\{z_\tau\}_{\tau=1}^{|\mathcal{V}|}$
        represents the labeling of the $\mathbf{v}_\tau$'s in $\mathcal{V}$
    /* M-step */
12     **for** $\{\mathcal{D}^s_\tau = \{(\mathbf{x}^s_i, y^s_i)\}_{i=1}^{ns}, \mathcal{D}^q_\tau = \{(\mathbf{x}^q_i, y^q_i)\}_{i=1}^{nq}\}$ *in Dataloader($\mathcal{D}^{tr}$)* **do**
          /* load a task episode */
13       $\{\mathbf{e}^s_i\}_{i=1}^{ns} = \{f_{\boldsymbol{\theta}}(\mathbf{x}^s_i)\}_{i=1}^{ns}$               // forward pass
14       $\{\mathbf{e}^q_i\}_{i=1}^{nq} = \{f_{\boldsymbol{\theta}}(\mathbf{x}^q_i)\}_{i=1}^{nq}$              // forward pass
15       $\{\boldsymbol{\mu}^c_1, ..., \boldsymbol{\mu}^c_N\}^s = $ Class-Pooling($\{(\mathbf{e}^s_i, y^s_i)\}_{i=1}^{ns}$)
16       $\boldsymbol{\mu}^a_{z_\tau} = $ Task-Pooling($\{\boldsymbol{\mu}^c_1, ..., \boldsymbol{\mu}^c_N\}^s$)       // the mean of $q_{\boldsymbol{\phi}}(\mathbf{v}_\tau|\mathcal{D}^s_\tau)$
17       Sample a task embedding $\mathbf{v}_\tau$ from $q_{\boldsymbol{\phi}}(\mathbf{v}_\tau|\mathcal{D}^s_\tau) = \mathcal{N}(\boldsymbol{\mu}^a_{z_\tau}, \bar{\sigma}^2\mathbf{I})$
18       **for** $j = 1, ..., N$ **do**
19         $\bar{\boldsymbol{\mu}}^c_j = \alpha\boldsymbol{\mu}^c_j + (1-\alpha)\mathbf{W}_{l^*}\mathbf{v}_{\tau'}$ where $l^* = \arg\min_{1\leq l\leq N} D(\boldsymbol{\mu}^c_j, \mathbf{W}_l\mathbf{v}_{\tau'})$
20       **end**
21       Calculate $\ell(\{\mathbf{e}^q_i\}_{i=1}^{nq}, \mathcal{V}, \{\bar{\boldsymbol{\mu}}^c_j\}_{j=1}^{N}, \{\boldsymbol{\mu}^t_1, ..., \boldsymbol{\mu}^t_r, \boldsymbol{\Sigma}^t_1, ..., \boldsymbol{\Sigma}^t_r\}, \sigma, \boldsymbol{\omega})$     // calculate the
          loss in Eq. (5) using Eq. (3) and Eq. (4)
22       $\boldsymbol{\theta}, \boldsymbol{\omega} = $ SGD($\ell, \boldsymbol{\theta}, \boldsymbol{\omega}$)           // update model parameters
23     **end**
24 **end**

---

## B.2   DISCUSSION ABOUT THE RELATED MULTI-TASK LEARNING METHODS

The modeling of the clustering/grouping structure of tasks or the mixture of distributions of tasks has been studied in multi-tasking learning (MTL). In (Xue et al., 2007; Jacob et al., 2008), tasks are assumed to have a clustering structure, and the model parameters of the tasks in the same cluster are drawn to each other via optimization on their L2 distances. In (Kang et al., 2011), a subspace based regularization framework was proposed for grouping task-specific model parameters, where the tasks in the same group are assumed to lie in the same low dimensional subspace for parameter sharing. The method in (Kumar & Daumé III, 2012) also uses the subspace based sharing of task parameters, but allows two tasks from different groups to overlap by having one or more bases in common. The method in (Passos et al., 2012) introduces a generative model for task-specific model parameters that encourages parameter sharing by modeling the latent mixture distribution of the parameters via the Dirichlet process and Beta process.

The key difference between these methods and our method HTGM lies in the difference between MTL and meta-learning. In an MTL method, all tasks are known *a priori*, *i.e.*, the testing tasks are from the set of training tasks, and the model is non-inductive at the task-level (but it is inductive at the sample-level). In HTGM, testing tasks can be disjoint from the set of training tasks, thus the model is inductive at the task-level. In particular, we aim to allow testing tasks that are not from the distribution of the training tasks by enabling the detection of novel tasks, which is an extension of the task-level inductive model. The second difference lies in the generative process. The method in (Passos et al., 2012) models the generative process of the task-specific model parameters (*e.g.*, the weights in a regressor). In contrast, HTGM models the generative process of each task by

generating the classes in it, and the samples in the classes hierarchically, *i.e.*, the $(\mathbf{x}, y)$'s (in Eq. (1) and Sec. 3.1). In this process, we allow our model to fit uniformly sampled classes given a task (without specifying a prior on the distance function on classes) by the proposed Gibbs distribution in Eq. (2). Other remarkable differences to the aforementioned MTL methods include the inference network (Fig. 1(b)), which allows the inductive inference on task embeddings and class prototypes; the optimization algorithm (Sec. 3.2) to our specific loss function in Eq. (3), which is from the likelihood in Eq. (1); and the model adaptation algorithm (Sec. 3.3) for performing predictions in a testing task, and detecting novel tasks. As such, the MTL methods can not be trivially applied to solve our problem.

### B.3 FURTHER INTERPRETATION OF THE TASK-CONDITIONAL DISTRIBUTION

The task-conditional class distribution $p_{\boldsymbol{\omega}}(y_i = k | \mathbf{v}_\tau)$ in Eq. (2) is defined through an energy function $E_{\boldsymbol{\omega}}(\boldsymbol{\mu}_k^c; \mathbf{v}_\tau) = \min\left(\{\|\boldsymbol{\mu}_k^c - \mathbf{W}_j\mathbf{v}_\tau\|_2^2\}_{j=1}^N\right)$ with trainable parameters $\boldsymbol{\omega} = \{\mathbf{W}_1, ..., \mathbf{W}_N\}$, for allowing uniformly sampled classes per task. The conditional distribution $p(y_i | \mathbf{v}_\tau)$ represents how classes distribute for a given task $\tau$. The reason for its definition in Eq. (2) is as follows. If it is a Gaussian distribution with $\mathbf{v}_\tau$ (*i.e.*, task embedding) as the mean, $p(y_i = k | \mathbf{v}_\tau)$ can be interpreted as the density at the representation of the $k$-th class in this Gaussian distribution, *i.e.*, the density at $\boldsymbol{\mu}_k$, which is the mean/surrogate embedding of the $k$-th class. One problem of this Gaussian $p(y_i | \mathbf{v}_\tau)$ is that different classes, *i.e.*, different $\boldsymbol{\mu}_{y_i}$'s, are not uniformly distributed, contradicting the practice that given a dataset (e.g., images), classes are often uniformly sampled for constituting a task in the empirical studies. Using a uniformly sampled set of classes to fit the Gaussian distribution $p(y_i | v_\tau)$ will lead to an ill-posed learning problem, as described in Sec. 3.1. To solve it, we introduced $\boldsymbol{\omega} = \{\mathbf{W}_1, ..., \mathbf{W}_N\}$ in the energy function $E_{\boldsymbol{\omega}}(\boldsymbol{\mu}_k^c; \mathbf{v}_\tau)$ in Eq. (2). $\mathbf{W}_j \in \mathbb{R}^{d \times d}$ $(1 \le j \le N)$ can be interpreted as projecting $\mathbf{v}_\tau$ to the $j$-th space spanned by the basis (*i.e.*, columns) of $\mathbf{W}_j$. There are $N$ different spaces for $j = 1, ..., N$. Thus, the $N$ projected task means $\mathbf{W}_1\mathbf{v}_\tau, ..., \mathbf{W}_N\mathbf{v}_\tau$ are in $N$ different spaces. Fitting the energy function $E_{\boldsymbol{\omega}}(\boldsymbol{\mu}_k^c; \mathbf{v}_\tau)$ to $N$ uniformly sampled classes $\boldsymbol{\mu}_1^c, ..., \boldsymbol{\mu}_N^c$, which tend to be far from each other because they are uniformly random, tends to learn $\mathbf{W}_1, ..., \mathbf{W}_N$ that project $\mathbf{v}_\tau$ to $N$ far apart spaces that fit each of the $\boldsymbol{\mu}_1^c, ..., \boldsymbol{\mu}_N^c$ by closeness, due to the min-pooling operation. This mitigates the aforementioned ill-posed learning problem.

## C APPENDIX FOR IMPLEMENTATION DETAILS

### C.1 THE SETUP OF THE COMPARED MODELS

**Encoder of Metric-based Meta-Learning.** For fairness, for all metric-based methods, including ProtoNet (Snell et al., 2017), MetaOptNet (Lee et al., 2019c), ProtoNet-Aug (Su et al., 2020), FEATS (Ye et al., 2020) and NCA (Laenen & Bertinetto, 2021), following (Tian et al., 2020; Lee et al., 2019c), we apply ResNet-12 as the encoder. ResNet-12 has 4 residual blocks, each has 3 convolutional layers with a kernel size of $3 \times 3$. ResNet-12 uses dropblock as a regularizer, and its number of filters is (60, 160, 320, 640). For MetaOptNet, following its paper (Lee et al., 2019c), we flattened the output of the last convolutional layer to acquire a 16000-dimensional feature as the image embedding. For other baselines, following (Tian et al., 2020), we used a global average-pooling layer on the top of the last residual block to acquire a 640-dimensional feature as the image embedding.

**Further Details.** Following (Snell et al., 2017), ProtoNet, ProtoNet-Aug, and NCA use Adam optimizer with $\beta_1 = 0.9$ and $\beta_2 = 0.99$. We did grid-search for the initial learning rate of the Adam within $\{1e^{-2}, 1e^{-3}, 1e^{-4}\}$, where $1e^{-3}$ was selected, which is the same as the official implementation provided by the authors. For FEATS, we chose transformer as the set-to-set function based on the results reported by (Ye et al., 2020). When pre-training the encoder in FEATS, following its paper (Ye et al., 2020), we applied the same setting as ProtoNet, which is to use Adam optimizer with an initial learning rate of $1e^{-3}$, $\beta_1 = 0.9$ and $\beta_2 = 0.99$. When training its aggregation function, we grid-searched the initial learning rate in $\{1e^{-4}, 5e^{-4}, 1e^{-5}\}$ since a larger learning rate leads to invalid results on our datasets. The optimal choice is $1e^{-4}$. For MetaOptNet, following its paper (Lee et al., 2019c), we used SGD with Nesterov momentum of 0.9, an initial learning rate of 0.1 and a scheduler to optimize it, and applied the quadratic programming solver OptNet (Amos & Kolter, 2017) for the SVM solution in it.

### C.2 THE DETAILS OF THE SETUP FOR NOVEL TASK DETECTION

In the experiments on novel task detection in Sec. 4.1, the number of in-distribution tasks (from the Original domain) in the test set is 4000 (1000 per task cluster) and the number of novel tasks (from the Blur and Pencil domains) in the test set is 8000 (4000 for the Blur and 4000 for the Pencil).

## D APPENDIX FOR EXPERIMENTAL RESULTS

### D.1 ANALYSIS OF $\sigma$

| Setting of $\sigma$ | Bird | Texture | Aircraft | Fungi |
|---|---|---|---|---|
| 0.1 | 69.33 | 46.92 | 75.20 | 50.78 |
| 0.5 | 70.00 | **47.98** | 75.38 | **52.38** |
| 1.0 (Ours) | **70.12** | 47.76 | **75.52** | 52.06 |
| 10.0 | 69.4 | 47.28 | 75.32 | 51.5 |

Table 4: Analysis of different $\sigma$

Tabel 4 report the effect of different $\sigma$ on the classification performance (5-way-1-shot classification on Multi-Plain dataset). As shown in the table, although the too low or too high setting of this hyper-parameter will hurt the performance, in general the model is robust toward the setting of $\sigma$.

### D.2 ANALYSIS OF $\bar{\sigma}$

| Setting of $\bar{\sigma}$ | Bird | Texture | Aircraft | Fungi |
|---|---|---|---|---|
| 0.05 | 69.78 | **48.36** | 74.36 | 51.34 |
| 0.1(Ours) | **70.12** | 47.76 | **75.52** | **52.06** |
| 0.2 | 70.02 | 47.50 | 75.30 | 51.74 |
| 0.5 | 69.02 | 46.66 | 74.46 | 51.00 |

Table 5: Analysis of different $\bar{\sigma}$

Tabel 5 summarize how different $\bar{\sigma}$ influence classification performance (5-way-1-shot classification on Multi-Plain dataset). In general, different settings of $\bar{\sigma}$ will influence the model performance at a marginal level, indicating our model's robustness toward this hyper-parameter.

### D.3 IMPACT OF GMM COMPONENT NUMBER

| Number of components $r$ | 2 | 4 | 8 | 16 | 32 |
|---|---|---|---|---|---|
| Silhouette score | 47.70 | 57.61 | 12.76 | 7.81 | 6.19 |

Table 6: Analysis on the number of mixture components

Different choices of the number of mixture components does not significantly influence the model classification performance. However, the clustering quality may vary due to the different numbers of components. Here, we report the Silhouette score (Shahapure & Nicholas, 2020; Sharma et al., 2021) *w.r.t.* the number in Table 6. From Table 6, we can see that selecting a component number close to the ground-truth component number of the distribution can benefit the clustering quality.

### D.4 CLASSIFICATION PERFORMANCE OF THE ABLATION VARIANTS

We summarize the classification performance of the two Ablation Variants HTGM w/o GMM and HTGM-Gaussian in Table 7. As we can see, our unique designs improve the novel task detection performance without significantly decreasing the classification performance.

| Ablation Variants | Bird | Texture | Aircraft | Fungi |
|---|---|---|---|---|
| HTGM w/o GMM | 68.86 | **48.00** | **75.74** | **52.28** |
| HTGM-Gaussian | 69.52 | 47.3 | 75.38 | 51.34 |
| HTGM | **70.12** | 47.76 | 75.52 | 52.06 |

Table 7: Ablation study of different variants of our proposed method.

| Setting | Model | Bird | Texture | Aircraft | Fungi | Average |
|---|---|---|---|---|---|---|
| 5-way-1-shot | ANIL-MAML | 62.64±0.90 | 43.86±0.78 | 70.03±0.85 | 48.34±0.89 | 56.22 |
| | ANIL-HSML | 64.33±0.87 | 43.77±0.79 | 69.71±0.84 | 47.75±0.89 | 56.39 |
| | ANIL-ARML | 65.98±0.87 | 43.57±0.78 | 70.28±0.84 | 48.48±0.92 | 57.08 |
| | HTGM (ours) | **70.12±1.28** | **47.76±1.49** | **75.52±1.24** | **52.06±1.41** | **61.37** |
| 5-way-5-shot | ANIL-MAML | 74.38±0.73 | 55.36±0.74 | 79.78±0.63 | 59.57±0.79 | 67.27 |
| | ANIL-HSML | 78.18±0.71 | 57.70±0.75 | 81.32±0.62 | 59.83±0.81 | 69.26 |
| | ANIL-ARML | 78.79±0.71 | 57.61±0.73 | 81.86±0.59 | 60.19±0.81 | 69.61 |
| | HTGM (ours) | **82.27±0.74** | **60.67±0.78** | **88.48±0.52** | **65.70±0.79** | **74.28** |

Table 8: More results (accuracy±95% confidence) of the optimization-based methods.

## D.5 ABLATION ANALYSIS OF OPTIMIZATION-BASED METHODS

Table 8 summarizes the performance of MAML, HSML and ARML trained in ANIL method (Raghu et al., 2020), *i.e.*, we pre-trained the ResNet-12 by ProtoNet, froze the encoder, and fine-tuned the last fully-connected layers using MAML, HSML and ARML on Plain-Multi dataset. From Table 8, the performance of ANIL-MAML is better than MAML in Table 1, similar to the observation in (Raghu et al., 2020), indicating the effectiveness of ANIL method. However, ANIL-HSML and ANIL-ARML perform similarly to ANIL-MAML, losing their superiority of modeling the mixture distribution of tasks achieved when implemented without ANIL as in Table 1 (up to 5.6% average improvement). This is because the cluster layer in HSML and the graph layer in ARML both affect the embeddings learned through backpropagation, *i.e.*, they were designed for joint training with the encoder. When the encoder is frozen, they cannot work properly. For this reason, to be consistent with the existing researches (Yao et al., 2019a;b) that demonstrated the difference between HSML/ARML and MAML, we used their original designs in Sec. 4. Meanwhile, we observe the proposed HTGM outperforms MAML, HSML, and ARML trained in ANIL method, this is because MAML cannot model the mixture distribution of tasks, while HSML and ARML cannot work properly when trained in ANIL method.

## D.6 LIMITATIONS OF THE PROPOSED METHOD

| Model | 5-way-1-shot | 5-way-5-shot |
|---|---|---|
| ProtoNet-Aug | 59.40±0.93 | 74.68±0.45 |
| HTGM | **61.80±0.95** | **74.55±0.45** |

Table 9: Comparison of our proposed method with other models on mini-imagenet dataset.

In the case when the task distribution is not a mixture, our model would degenerate to and perform similarly to the general metric-based meta-learning methods, *e.g.*, ProtoNet, which only considers a uni-component distribution. To confirm this, we added an experiment that compares our model with ProtoNet-Aug on Mini-Imagenet (Vinyals et al., 2016), which does not have the same explicit mixture distributions as in the Plain-Multi and Art-Multi datasets in Section 4. The results are summarized in Table 9. From the table, we observe our method performs comparably to ProtoNet, which validates the aforementioned guess. Meanwhile, together with the results in Table 1 and Table 2, the proposed method could be considered as a generalization of the metric-based methods to the mixture of task distributions.

