# OpenReview forum: "Hierarchical Gaussian Mixture based Task Generative Model for Robust Meta-Learning"
_ICLR.cc/2023/Conference — Submitted to ICLR 2023_

### Official Review · Reviewer_DtsZ · 2022-10-22

**Confidence:** 4
**Correctness:** 3
**Technical Novelty And Significance:** 2
**Empirical Novelty And Significance:** 3
**Recommendation:** 5

**Clarity, Quality, Novelty And Reproducibility:**

The paper is overall clearly written

The proposed method is not new itself.

The results seem to be reproducible.

**Strength And Weaknesses:**

**Strengths**:

* The proposed method can handle both a mixture of task distributions and detecting novelty in testing tasks is interesting.


**Weakness**:

* The proposed method is an application of Hierarchical Gaussian Mixture to the meta learning setting. The method itself is not novel and is a well-studied model in the machine learning community.


* For the novel task detection experiments, it would be better to include more experiments that use Mini-ImageNet as source, and  CIFAR, aircraft, etc as novel tasks datasets to see whether it works well in practice.


* The compared methods of optimization-based methods appear difficult to scale to a large backbone, but it is not always the case. For example, you can pre-train the backbone and fix most of layers, and only train the last or last few layers, similar to ANIL [1]

References;

[1] Rapid Learning or Feature Reuse? Towards Understanding the Effectiveness of MAML. ICLR 2020


**Summary Of The Paper:**

This paper proposes a Hierarchical Gaussian Mixture method to parametrize the generation process of tasks in meta-learning.  The proposed model can be used for fitting a mixture of task distributions and evaluating the scoring of testing tasks for evaluating the novelty of the testing task.  Experiments on several datasets demonstrate the effectiveness of the proposed methods.

**Summary Of The Review:**

This paper proposes a Hierarchical Gaussian Mixture method to parametrize the generation process of tasks in meta-learning. The method itself is not new and experiments need to be further improved.

---

> ### Author Response · Authors · 2022-11-19
> **Response to the comments from the reviewer (part 1)**
>
> Thank you so much for the constructive feedback. We sincerely appreciate your valuable suggestions and questions. The following are our responses.
>
> Q1. The proposed method is an application of Hierarchical Gaussian Mixture to the meta learning setting. The method itself is not novel and is a well-studied model in the machine learning community.
>
> Thanks for the careful attention to the traditional works on Hierarchical Gaussian Mixture model. We also noticed the relevant works, and discussed them briefly in the related work (in the last paragraph in Section 2). To the best of our knowledge, the term Hierarchical Gaussian Mixture (HGM) model has mostly appeared in the traditional works (Goldberger & Roweis, 2005; Olech & Paradowski, 2016; Athey et al., 2019) for hierarchical clustering by applying Gaussian Mixture model agglomeratively or divisively on the input vectorial samples. They are unsupervised methods that infer clusters of samples, but do not pre-train embedding models (or parameter initializations) that could be fine-tuned for the adaptation to new tasks in meta-learning. Therefore, these methods are remarkably different from meta-learning methods, and we think it is a non-trivial problem to adapt the concept of HGM to solve the meta-learning problem. To this end, we need to (1) identify the right motivation; and (2) solve the new technical challenges. For (1), we found the hierarchical structure of mixture distributions naturally appears when we want to model the generative process of tasks from a mixture of distributions, where each task contains another mixture distribution of classes (as suggested by Eq. (1)). In other words, the motivating point of our method is more on meta-learning than the HGM. However, we think drawing such a connection between meta-learning and HGM is a novel contribution. For (2), our method is different from traditional HGM in
>
> (a) its generative process of tasks (page 4), which is a theoretical extension of the widely used empirical process of generating tasks in meta-learning;
>
> (b) its Gibbs-style task-conditional distribution (Eq. (2)) for fitting uniformly sampled classes;
>
> (c) the metric-based end-to-end meta-learning framework (Figure 1) (note the traditional HGM is not for learning embeddings);
>
> (d) the derivation of the optimization algorithm to the proposed learning problem in Section 3.2 and Algorithm 1 (in Appendix A.4);
>
> (e) the corresponding novel model adaptation process in Section 3.3.
>
> We think solving the technical challenges in the proposed new generative model is another novel contribution. We appreciate this question and agree that the difference should be more detailed. Therefore, we added the aforementioned descriptions in Appendix B.1, and referred to it in the last paragraph in Section 2 (Related Work). We also added a description about the key differences in the last paragraph in Section 2. Please let us know should we overlooked any related works, and we appreciate the opportunity to become more comprehensive in the discussion on the contributions.
>
> Q2. For the novel task detection experiments, it would be better to include more experiments that use Mini-ImageNet as source, and CIFAR, aircraft, etc as novel tasks datasets to see whether it works well in practice.
>
> Thank you for the suggestion. In this paper, we focus on the meta-learning on tasks from a mixture of distributions. However, the commonly used uni-component datasets like Mini-ImageNet and Aircraft do not explicitly provide such mixture property for evaluating methods that model the mixture distribution of tasks. As a result, detecting novel tasks with respect to such uni-component datasets may not be as challenging as our case. To indicate it, we performed an experiment that used Mini-ImageNet as the source and used FGVC of the Aircraft dataset as the novel task dataset. The result is shown in the following table. As we can see, our proposed method outperforms its uni-component distribution based counterpart ProtoNet-Aug, but the boost is less remarkable compared to the mixture distribution setting (in Table 3) because a simple baseline is already good enough, leaving a small room for improvement. For example, in the table, ProtoNet-Aug achieved a high performance with AUROC of 94.92% and Average Precision of 93.73%.
> |Model        | AUROC |  AP   | Max-F1 |
> |-------------|-------|-------|--------|
> |ProtoNet-Aug | 94.92 | 93.73 | 88.67  |
> |HTGM(ours)         | **97.66** | **97.54** | **92.49**  |

---

> ### Author Response · Authors · 2022-11-19
> **Response to the comments from the reviewer (part 2)**
>
> Q3. The compared methods of optimization-based methods appear difficult to scale to a large backbone, but it is not always the case. For example, you can pre-train the backbone and fix most of layers, and only train the last or last few layers, similar to ANIL [1] ([1] Rapid Learning or Feature Reuse? Towards Understanding the Effectiveness of MAML. ICLR 2020)
>
> We are thankful to this suggestion. Following the suggestion, we used the training algorithm in ANIL to reimplement MAML (the most representative optimization-based method), HSML and ARML (the two best optimization-based baselines in our experiments). Specifically, we froze the encoder that was pre-trained with ProtoNet, and only fine-tuned the other layers of the model of MAML, HSML and ARML, respectively, for ablation study on the Plain-Multi dataset. The experimental results are included in Appendix D.5 in the revised draft, and was referred to in the "Implementation" section in Section 4 (above Section 4.1).
>
> The results are copied in the following.
>
> |Setting     |Model     |      Bird      |   Texture  |    Aircraft    |    Fungi   | Average |
> |------------|----------|----------------|------------|----------------|------------|---------|
> |            |ANIL-MAML |   62.64±0.90   | 43.86±0.78 |   70.03±0.85   | 48.34±0.89 |  56.22  |
> |5-way-1-shot|ANIL-HSML |   64.33±0.87   | 43.77±0.79  |   69.71±0.84   | 47.75±0.89 |  56.39  |
> |            |ANIL-ARML |   65.98±0.87   | 43.57±0.78  |   70.28±0.84   | 48.48±0.92 |  57.08  |
> |            |HTGM(ours)| **70.12±1.28**|**47.76±1.49**|**75.52±1.24**| **52.06±1.41** | **61.37** |
>
> |Setting     |Model     |      Bird      |   Texture  |    Aircraft    |    Fungi   | Average |
> |------------|----------|----------------|--------------|----------------|------------|---------|
> |            |ANIL-MAML |   74.38±0.73   | 55.36±0.74 |   79.78±0.63   | 59.57±0.79 |  67.27  |
> |5-way-5-shot|ANIL-HSML |   78.18±0.71   | 57.70±0.75 |   81.32±0.62   | 59.83±0.81 |  69.26  |
> |            |ANIL-ARML |   78.79±0.71   | 57.61±0.73 |   81.86±0.59   | 60.19±0.81 |  69.61  |
> |            |HTGM(ours)| **82.27±0.74** | **60.67±0.78** | **88.48±0.52** | **65.70±0.79** | **74.28** |
>
> First, we observe our proposed method outperforms MAML, HSML, and ARML trained in ANIL method, this is because MAML cannot model the mixture distribution of tasks, while HSML and ARML cannot work properly when trained in ANIL method (the details are in the following).
>
> The performance of ANIL-MAML is better than MAML in Table 1, similar to the observation in (Raghu et al., 2020), indicating the effectiveness of ANIL method. However, ANIL-HSML and ANIL-ARML perform similarly to ANIL-MAML, losing their superiority of modeling the mixture distribution of tasks achieved when implemented without ANIL as in Table 1 (up to 6.08% average improvement). This is because the cluster layers in HSML and the graph layers in ARML both affect the embeddings learned through backpropagation, i.e., they were designed for joint training with the encoder. When the encoder is frozen, they cannot work properly. For this reason, to be consistent with the existing works (Yao et al., 2019a, 2019b) that demonstrated the differences between HSML/ARML and MAML, we used their original designs in Table 1 and Table 2, and reported the ablation study with ANIL in Appendix D.5.

---

> ### Author Response · Authors · 2022-12-08
> **Looking forward to hearing from you**
>
> Dear Reviewer,
>
> Thank you again for your insightful comments. Following your suggestions, we included new experimental results, and clarified the novelty of the proposed method and the distinction between it and the conventional Hierarchical Gaussian Mixture model. We would be very grateful if we could hear from you to see if our response helps address the questions, and take the opportunity before the end of the discussion period to address any further questions. Thanks!

---

> ### Author Response · Authors · 2022-12-10
> **Really appreciate your comment and looking forward to hearing from you**
>
> Dear Reviewer,
>
> We sincerely appreciate your valuable comments on our work. We recently added a summary of the revisions to the paper in a comment for potential help with the review of the revised paper. In our previous response and the revised paper, we tried our best to address the questions and take the suggestions in your review, which are very helpful for us to improve our work. Also, we are wondering if our previous response can help justify our work, and should we clarify any further questions?
>
> Thank you!

---

> ### Author Response · Authors · 2022-12-12
> **Follow up: looking forward to hearing from you**
>
> Dear Reviewer,
>
> We are grateful to your review. As it is approaching to the end of the discussion period, can we hear your thoughts on our response to your questions and the revised paper, and on how could we do for any further questions?
>
> Thank you for your attention!

---

### Official Review · Reviewer_gaxY · 2022-10-30

**Confidence:** 3
**Correctness:** 3
**Technical Novelty And Significance:** 3
**Empirical Novelty And Significance:** 2
**Recommendation:** 6

**Clarity, Quality, Novelty And Reproducibility:**

This approach to task-modeling in few-shot learning is an interesting idea.  The descriptions are clear with a reasonable amount of attention and study, but I think they could be simplified or made more consistent (see comments above).

**Strength And Weaknesses:**

Modeling task distribution in this way is an interesting idea, and the results of the system are promising, improving upon appropriate baselines.  The formulation is rather complex, leading to a somewhat complicated system, and whether the improvements are worth the complexity is debatable -- but it's certainly a reasonable trade-off.

I think some of the notation and descriptions could be made simpler (see some suggestions below), which might help (see below comments), but the system is pretty described as it is.  I had some trouble understanding the inference procedure, though:  In sec 3, I didn't understand the process for inferring the task embedding $v_{\tau'}$.  This section says a single $v$ is sampled and then means are adapted to drawn towards the Wv means; but should this use either the most likely $v$ given class prototypes $\mu^s$, or a larger sample of $v$ from the distribution?

Also, another approach might have been to use backpropagation, rather than EM, perhaps combined with Gumbel-softmax on the categorical from a direct weight parameterization.  Have you explored other potentially simpler approaches (I'm not sure if this one would work) that still capture the task distribution model?

Overall this is an interesting idea and model, with reasonable effectiveness.  It is somewhat complex, though, and I wonder if it (or some of its descriptions) could be simplified.




Further questions and comments:

The notation is a little involved and can be hard to follow sometimes, e.g. differences between $\mu$, $\hat\mu$, $\bar\mu$.  Since all the distributions are MOG or MOG-like, except the top level categorical, maybe it could work to index each level instead, e.g. calling the means $\mu^l$ where $l$ is the hierarchy level, and similarly for other variables.

Is there some intuition for this form of W parameterization, for how it models a task distribution?  it looks like a dictionary W * coeffs v, is one interpretation?  In the limiting case it might have a blockwise W_i = [0 | w_i1, ..., w_ik | 0], with w_ik being a basis for task i, and coeffs in the corresponding block of v.  The actual system of course is not constrained to this --- but is there a similar way to look at this with subspaces, and are the subspaces "tied together" between tasks, so that instead of forming a basis of each class mean they are forming one of task space?

eq 2 Gibbs distribution is basically another mixture of gaussians, with means {W_j v}.  Why not use MOG here exactly?

challenges 1, 2:  To me, the l_neg sampling in challenge 1 looks like the contrastive term that would be supplied by partition function mentioned in challenge 2 --- certainly it's of the same form and is subtracted.  I'm not entirely sure of its exact placement, whether it's a substitute for this or if they are in separate places.  Do they correspond in this way, and if so, would the trivial solution in challenge 1 come up if the partition function weren't replaced with a constant bound?

Text in 3.3 uses $\psi$ referring to Fig 1b, but Fig 1b doesn't mention $\psi$ anywhere in it.  In general there is a bit of back-and-forth between use of $\psi$ and $\theta$, since it's mentioned $\psi = \theta$.  Could this be made more consistent?

Sec 4.1:  The ProtoNet + GMM baseline is good, simple and very appropriate.  How many GMM components were used, and how large was the in-distribution training sample?  Were these tuned in any way to best performance for this baseline approach?


**Summary Of The Paper:**

This paper describes an approach for few-shot learning, that provides a way to model not only class distributions, but the distribution of tasks.  This provides two benefits: (i) improved performance on few-shot tasks, and more interestingly, (ii) the ability to determine whether a task (support set of (images, class) pairs) is "novel" and too far outside the domains the few-shot learner was trained on.  While other works always evaluate on novel classes, sometimes chosen to come from semantically different subtrees than those in meta-training, this work explicitly models the distribution of tasks themselves using Gibbs distribution with basis {W}, in order to identify out-of-distribution tasks.  This forms a hierarchy of mixture models, describing the tasks, classes and embeddings.  The system is optimized using EM, with SGD in the M step.  It is evaluated on Plain-Multi and Art-Multi datasets, finding improved performance over current few-shot-learning techniques augmented with GMM in-distribution model.


**Summary Of The Review:**

Overall this is an interesting idea and model, with reasonable effectiveness.  It is somewhat complex, though, and I wonder if it (or some of its descriptions) could be simplified.

---

> ### Author Response · Authors · 2022-11-19
> **Response to the comments from the reviewer (part 1)**
>
> Thank you so much for the constructive feedback. We sincerely appreciate your valuable suggestions and questions. The following are our responses.
>
> Q1. In sec 3, I didn't understand the process for inferring the task embedding $v_{\tau'}$. This section says a single $v$ is sampled and then means are adapted to drawn towards the Wv means; but should this use either the most likely v given class prototypes $\mu^{s}$, or a larger sample of $v$ from the distribution?
>
> Thanks for the insightful question. We think it is about the model adaptation in Section 3.3. We agree with the reviewer that in this process, the most likely $v_{\tau'}$ should be used instead of a single sample. We are now aware that the phrase of the sentence "(2) distribution $q_{\boldsymbol{\phi}}(v_{\tau'}|D_{v_{\tau'}}^{s})$ , from which we sample a task embedding $v_{\tau'}$." may not be rigorous. In fact, we used the mean of $q_{\boldsymbol{\phi}}(v_{\tau'} | D_{v_{\tau'}}^{s})$, i.e., $\boldsymbol{\mu_{z_{\tau'}}}$, directly to replace $v_{\tau'}$ in this step in our implementation. In the previous draft, we tried to indicate this in Figure 1(b) by drawing the blue arrow (i.e., sampling) directly from the red star (i.e., the mean). In the revised draft, we changed the sentence to clarify it. Now it becomes "(2) distribution $q_{\boldsymbol{\phi}}(v_{\tau'} | D_{v_{\tau'}}^{s})$, from which we draw the average task embedding $v_{\tau'}=\boldsymbol{\mu_{z_{\tau'}}}^{a}$.".
>
> Q2. Another approach might have been to use backpropagation, rather than EM, perhaps combined with Gumbel-softmax on the categorical from a direct weight parameterization. Have you explored other potentially simpler approaches (I'm not sure if this one would work) that still capture the task distribution model?
>
> Yes, in our experiments, we tried to use the backpropagation to directly optimize the objective function (Eq.(5)). However, an issue is that, the GMM model may degenerate to a trivial solution which assign all tasks into one component (similarly observed by Dias, José G., and Michel Wedel, 2004 [1] ). This is because assigning all tasks into one component is actually a local (but not global) optimal solution that can be easily achieved by the SGD algorithm. In contrast, in EM algorithm the posterior distribution $p(z_{\tau})$ inferred in E-step will be frozen in the M-step and thus regularize the SGD algorithm to disperse the tasks according to $p(z_{\tau})$.
>
> * [1] Dias, José G., and Michel Wedel. "An empirical comparison of EM, SEM and MCMC performance for problematic Gaussian mixture likelihoods." Statistics and Computing 14.4 (2004): 323-332.
>
> Q3. The notation is a little involved and can be hard to follow sometimes, e.g. differences between $\mu$, $\hat{\mu}$, $\bar{\mu}$. Perhaps it could work to index each level instead, e.g. calling the means $\mu^{l}$ where $l$ is the hierarchy level, and similarly for other variables.
>
> Thanks for the suggestion. We made the following changes for improving the readability of the notations.
> * $\boldsymbol{\mu} \rightarrow \boldsymbol{\mu}^{c}$ to denote the class-level means.
> * $\boldsymbol{\Sigma} \rightarrow \boldsymbol{\Sigma}^{c}$ to denote the class-level variance.
> * $\boldsymbol{\bar{\mu}} \rightarrow \boldsymbol{\mu}^{t}$ to denote the task-level means.
> * $\boldsymbol{\bar{\Sigma}} \rightarrow \boldsymbol{\Sigma}^{t}$ to denote the task-level variance.
> * $\boldsymbol{\hat{\mu}} \rightarrow \boldsymbol{\mu}^{a}$ to denote the means inferred by the inference network in Section 3.2.
>
> We used $\sigma$ for the isotropic Gaussian with tied variance $\boldsymbol{\Sigma_1}^{c}=...=\boldsymbol{\Sigma_N}^{c}=\sigma^{2}\textbf{I}$. We didn't add superscript $c$ to $\sigma$ because it is squared with another superscript 2. Similarly, we used $\bar{\sigma}$ for the variance of the approximated posterior $q_{\boldsymbol{\phi}}$. Meanwhile, to distinguish the superscripts $c$, $t$, $a$ from the power operation, we used a different font for them (sans serif) in the revised draft. We also updated the notations in Figure 1, where we used $\\{\boldsymbol{\mu_{1}}^{c}, ..., \boldsymbol{\mu_{1}}^{c}\\}^{s}$ and $\\{\boldsymbol{\mu_{1}}^{c}, ..., \boldsymbol{\mu}_{1}^{c}\\}^{q}$ to distinguish the class-level means from the support set and query set, respectively.

---

> ### Author Response · Authors · 2022-11-19
> **Response to the comments from the reviewer (part 2)**
>
> Q4. Is there some intuition for this form of W parameterization, for how it models a task distribution? It looks like a dictionary W * coeffs v, is one interpretation?
>
> This is an interesting perspective of interpretation. Let us elaborate the reason for introducing $W$'s.
>
> $W_{1}, ..., W_{N}$ were introduced in the definition of $p(y_{i}|v_{\tau})$ in Eq. (2), which represents how classes distribute for a given task $\tau$. If it is a Gaussian distribution with $v_{\tau}$ (task embedding) as the mean, $p(y_{i}=k|v_{\tau})$ can be interpreted as the density at the representation of the $k$-th class in this Gaussian distribution, i.e., the density at $\mu_{k}$ (the mean/surrogate embedding of the $k$-th class). One problem of this Gaussian-style $p(y_{i}|v_{\tau})$ is that different classes, i.e., different $\mu_{y_{i}}$'s, are not uniformly distributed, contradicting the practice that given a dataset (e.g., images), people often sample classes uniformly for constituting a task in the empirical studies. Using a uniformly sampled set of classes to fit the Gaussian distribution $p(y_{i}|v_{\tau})$ will lead to an ill-posed learning problem. To solve it, we introduced $W_{1}, ..., W_{N}$ in the energy function $E(\mu_{k}; v_{\tau})=\min(\\{||\mu_k - W_j v_\tau||^2_2\\}^N_{j=1})$ for the Gibbs distribution $\pi$. Here, $W_{j}\in\mathbb{R}^{d\times d}$ can be interpreted as projecting $v_{\tau}$ to the $j$-th space spanned by the basis (i.e., columns) of $W_{j}$. There are N different spaces for $j=1, ..., N$. Thus, the N projected task means $W_{1}v_{\tau}, ..., W_{N}v_{\tau}$ are in N different spaces. Fitting the energy function $E$ to N uniformly sampled classes $\mu_{1}^{c}, ..., \mu_{N}^{c}$, which tend to be far from each other because they are uniformly random, tends to learn $W_{1}, ..., W_{N}$ that project $v_{\tau}$ to N far apart spaces that fit each of the $\mu_{1}^{c}, ..., \mu_{N}^{c}$ by closeness, due to the min-pooling operation. This mitigates the aforementioned ill-posed learning problem. We added this interpretation in Appendix B.3, and refered to it at the end of the "Task-Conditional Distribution" section in Section 3.1.
>
> Q5. Eq 2 Gibbs distribution is basically another mixture of gaussians, with means {W_j v}. Why not use MOG here exactly?
>
> The energy function $E(\mu_{k}; v_{\tau})=\min(\\{||\mu_k - W_j v_\tau||^2_2\\}^N_{j=1})$ in Eq. (2) contains N L2 norms, which basically conveys N Gaussian distributions with means $W_{1}v_{\tau}, ..., W_{N}v_{\tau}$, respectively. The difference between it and a conventional mixture of Gaussian distributions lies in the min-pooling operation, which is important for enabling N equal (global) optimums for uniformly sampled classes as discussed in Q4. In MOG, usually the N Gaussian distributions are weighted (by the probabilities of the N different components) and summed up, which cannot achieve the N equal (global) optimums as we desired in Q4.
>
> Q6. Challenges 1, 2: the l_neg sampling in challenge 1 looks like the contrastive term that would be supplied by partition function mentioned in challenge 2. Do they correspond in this way, and if so, would the trivial solution in challenge 1 come up if the partition function weren't replaced with a constant bound?
>
> Thanks for the insightful question. They are not exactly corresponded, but we agree that the partition function could be helpful for avoiding the trivial solution if it weren't replaced with a constant bound. The partition function helps avoid the case when the class means fall into the same point. However, since it involves an integral, it is hard to be efficiently computed. In light of this, we proposed a more tractable l_neg sampling to handle challenge 1, which helps avoid the trivial solution from the perspective of the sample embeddings.

---

> ### Author Response · Authors · 2022-11-19
> **Response to the comments from the reviewer (part 3)**
>
> Q7. Text in 3.3 uses $\psi$ referring to Fig 1b, but Fig 1b doesn't mention $\psi$ anywhere in it. In general there is a bit of back-and-forth between use of $\psi$ and $\theta$, since it's mentioned $\psi=\theta$. Could this be made more consistent?
>
> Before revising $\psi$, we noticed we can simplify the use the symbols by removing $\phi$ because $\mu_{1}^{t}, ..., \mu_{r}^{t}$ (which were originally included in $\phi$) are not part of the parameters solved by SGD. Instead, they are solved by the GMM inference as indicated in Algorithm 1. Thus the model parameters solved by SGD are $\theta$ and $\omega$, and the symbol $\phi$ can be ignored. As such, we replace $\psi$ with $\phi$, and in the revised draft, $\phi$ represents the inference network parameters.
>
> Then, we updated Figrue 1(a) by adding $q_{\boldsymbol{\phi}}(v_\tau | D^s_\tau)$ after "Task-pooling", and added $\mu_{z_{\tau}}^{a}$ close to the "star" symbol (i.e., task mean) below it. We also updated Figure 1(b) by adding $q_{\boldsymbol{\phi}}(v_\tau | D^s_\tau)$ after "Task-pooling", and added $\mu_{z_{\tau'}}^{a}$ close to the "star" symbol (i.e., task mean) below it.
>
> For consistency of notations, we tried to substitute all $\phi$ with $\theta$ after stating $\phi=\theta$ in the "Inference Network" subsection of Section 3.2, but later found this could lead to some confusion when readers look back to Eq. (3) where $\phi$ was used before introducing the details of the inference network. Since Section 3.3 is the only place that refers to $\phi$ after the "Inference Network" subsection, to avoid back-and-forth reference, in the fourth line of Section 3.3, we added "Recall that the inference network is the base model $f_{\boldsymbol{\theta}}(\cdot)$ with class-pooling and task-pooling layers, as illustrated in Figure 1(b), and $\boldsymbol{\phi}=\boldsymbol{\theta}$." to re-elaborate the concept.
>
> Q8. Sec 4.1: The ProtoNet + GMM baseline is good, simple and very appropriate. How many GMM components were used, and how large was the in-distribution training sample? Were these tuned in any way to best performance for this baseline approach?
>
> The ProtoNet+GMM baseline was used in the evaluation of novel task detection. Similar to the number of mixture components in the proposed HTGM, the number of GMM components for ProtoNet was grid-searched in [2, 4, 8, 16, 32], and 4 was selected. The number of in-distribution tasks (from the Original domain) in the test set is 4000 (1000 per mixture component of the tasks) and the number of novel tasks (from the Blur and Pencil domains) in the test set is 8000 (4000 for the Blur and 4000 for the Pencil). These details were added to Appendix C.2 and referred to in Section 4.1, "novel task detection", the 2nd paragraph. We also revised Appendix C.1 to elaborate the configuration setup and the selection method of hyperparameters for each of the baseline methods.

---

> ### Author Response · Authors · 2022-12-12
> **Thank you and looking forward to hearing from you**
>
> Dear Reviewer,
>
> Thank you for your insightful comments. Following your valuable suggestions, we revised and simplified the notations, included more discussions of the method, parameters, and experimental setups. Your suggestions are very helpful for us to improve our work, and we would be very grateful if we could hear from you to see if our response helps address the questions, and take the opportunity before the end of the discussion period to address any further questions.
>
> We also added a summary of the revisions to the paper in a comment above for potential help with the review of the revised paper.
>
> Thank you for your attention!

---

### Official Review · Reviewer_A867 · 2022-10-31

**Confidence:** 4
**Correctness:** 4
**Technical Novelty And Significance:** 3
**Empirical Novelty And Significance:** 3
**Recommendation:** 6

**Clarity, Quality, Novelty And Reproducibility:**

Could you please discuss the limitation or in which case the algorithm will fail?

**Details Of Ethics Concerns:**

N.A.

**Strength And Weaknesses:**

Strength

S1. I like the idea. Based on the observation that embedding is the key of few-shot learning, modeling with mixture of Gaussian makes perfect sense and is mathmatically solvable.

S2. The paper is well written and easy to follow.

Weaknesses

W1. Task grouping was extensively studied in the context of multi-task learning, e.g., [R1]. As both such mult-task learning algorithms (with metric-based few-shot algorithm pluggined) and the proposed method address a similar problem, agnostic to feature extractors, I think it is worthy discussing them, especially those with similar mathematical tools, in Related works.

[R1] Flexible Modeling of Latent Task Structures in Multitask Learning. ICML 2012.

W2. Experimental comparisons could be more solid. E.g.,  to compare with baselines also designed for dealing with mixture distributions of tasks, i.e., HSML and ARML, both the backbone and base algorithm are different, making it less convincing that the proposed method is better.

**Summary Of The Paper:**

This paper proposed a meta-training framework underlain by a novel Hierarchical Gaussian Mixture based Task Generative Model (HTGM). The basic assumption is that task embedding fits mixture distribution of Gaussian. The model parameters are learned end-to-end by maximum likelihood estimation via an Expectation-Maximization algorithm. Extensive experiments on benchmark datasets indicated the effectiveness of the method for both sample classification and novel task detection.

**Summary Of The Review:**

This paper proposed a novel algorithm for meta learning. It is based on embedding-based few shot learning and mixture distribution of Gaussian for task generation. The experiments also demonstrated the effectiveness of few-shot learning and novel task detection. Overall the paper is well written.

---

> ### Author Response · Authors · 2022-11-19
> **Response to the comments from the reviewer (part 1)**
>
> Thank you so much for the constructive feedback. We sincerely appreciate your valuable suggestions and questions. The following are our responses.
>
> Q1. Task grouping was extensively studied in the context of multi-task learning, it is worthy to discuss them, especially those with similar mathematical tools, in Related works.
>
> Thanks for drawing our attention to the related works in multi-task learning (MTL). We read the suggested paper (Passos et al. "Flexible Modeling of Latent Task Structures in Multitask Learning." In ICML, 2012.), and see the relevance, i.e., the modeling of the latent mixture of structures of tasks.
>
> First, please allow us to discuss the key difference between this method and our method, which lies in the difference between MTL and meta-learning. In the MTL method (Passos et al., 2012), all tasks are known a priori, i.e., the testing tasks are from the set of training tasks, and the model is non-inductive at the task-level (but it is inductive at the sample-level). In our method, testing tasks can be disjoint from the set of training tasks, thus the model is inductive at the task-level. In particular, we aim to allow testing tasks that are not from the distribution of the training tasks by enabling the detection of novel tasks, which is an extension of the task-level inductive model. The second difference lies in the generative process. The method in (Passos et al., 2012) models the generative process of the task-specific model parameters (e.g., the weights in a regressor). In contrast, our method models the generative process of each task by generating the classes in it, and the samples in the classes hierarchically, i.e., the (x, y)'s (in Eq. (1) and page 4). In this process, we allow our model to fit uniformly sampled classes given a task (without specifying a prior on the distance function on classes) by the proposed Gibbs distribution in Eq. (2). Other remarkable differences include the inference network (Figure 1(b)), which allows the inductive inference on task embeddings and class prototypes; the optimization algorithm (Section 3.2) to our specific loss function in Eq. (3), which is from the likelihood in Eq. (1); and the model adaptation algorithm (Section 3.3) for performing predictions in a testing task, and detecting novel tasks. As such, the MTL method (Passos et al., 2012) can not be trivially applied to solve our problem.
>
> Meanwhile, we agree this work should be discussed in the related works. We included following papers on task clustering/grouping in MTL in our discussion, which was put in Appendix B.2, and referred to it in the last paragraph in Section 2 (Related Work) by adding "Moreover, we discuss the relevant multi-task learning methods with task grouping in Appendix B.2.". The differences between these methods and our method are similar as in the above discussions, and the details are in Appendix B.2.
> * Xue et al. "Multi-Task Learning for Classification with Dirichlet Process Priors."  J. Mach. Learn. Res., 2007.
> * Jacob et al. "Clustered Multi-task Learning: a Convex Formulation." In NIPS, 2008.
> * Kang et al. "Learning with Whom to Share in Multi-task Feature Learning." In ICML, 2011.
> * Kumar and Daume III. "Learning Task Grouping and Overlap in Multi-Task Learning." In ICML, 2012.
> * Passos et al. "Flexible Modeling of Latent Task Structures in Multitask Learning." In ICML, 2012.

---

> ### Author Response · Authors · 2022-11-19
> **Response to the comments from the reviewer (part 2)**
>
> Q2. Experimental comparisons could be more solid. E.g., to compare with baselines also designed for dealing with mixture distributions of tasks, i.e., HSML and ARML, both the backbone and base algorithm are different, making it less convincing that the proposed method is better.
>
> We are thankful to this comment. Directly replacing the backbone in HSML/ARML with the same backbone as our model (i.e., ResNet) is intractable, because the optimization-based methods, such as HSML and ARML, were built upon MAML, which is difficult to scale to large backbones due to the high memory and time cost of the nested loops during training. In our experiments, running MAML, HSML and ARML on a 20GB NVIDIA RTX-3080 GPU reports out-of-memory errors (all methods in Section 4 used the same machine). To address this issue, following ANIL (Raghu et al. "Rapid Learning or Feature Reuse? Towards Understanding the Effectiveness of MAML." In ICLR, 2020.), we pretrained the ResNet-12 by ProtoNet, froze the encoder, and fine-tuned the other layers using MAML, HSML and ARML for an ablation study on the Plain-Multi dataset. The experimental results are included in Appendix D.5 in the revised draft, and was referred to in the "Implementation" section in Section 4 (above Section 4.1).
>
> The results are copied in the following.
>
> |Setting     |Model     |      Bird      |   Texture  |    Aircraft    |    Fungi   | Average |
> |------------|----------|----------------|------------|----------------|------------|---------|
> |            |ANIL-MAML |   62.64±0.90   | 43.86±0.78 |   70.03±0.85   | 48.34±0.89 |  56.22  |
> |5-way-1-shot|ANIL-HSML |   64.33±0.87   | 43.77±0.79  |   69.71±0.84   | 47.75±0.89 |  56.39  |
> |            |ANIL-ARML |   65.98±0.87   | 43.57±0.78  |   70.28±0.84   | 48.48±0.92 |  57.08  |
> |            |HTGM(ours)| **70.12±1.28**|**47.76±1.49**|**75.52±1.24**| **52.06±1.41** | **61.37** |
>
> |Setting     |Model     |      Bird      |   Texture  |    Aircraft    |    Fungi   | Average |
> |------------|----------|----------------|--------------|----------------|------------|---------|
> |            |ANIL-MAML |   74.38±0.73   | 55.36±0.74 |   79.78±0.63   | 59.57±0.79 |  67.27  |
> |5-way-5-shot|ANIL-HSML |   78.18±0.71   | 57.70±0.75 |   81.32±0.62   | 59.83±0.81 |  69.26  |
> |            |ANIL-ARML |   78.79±0.71   | 57.61±0.73 |   81.86±0.59   | 60.19±0.81 |  69.61  |
> |            |HTGM(ours)| **82.27±0.74** | **60.67±0.78** | **88.48±0.52** | **65.70±0.79** | **74.28** |
>
> First, we observe our proposed method outperforms MAML, HSML, and ARML trained in ANIL method, this is because MAML cannot model the mixture distribution of tasks, while HSML and ARML cannot work properly when trained in ANIL method (the details are in the following).
>
> The performance of ANIL-MAML is better than MAML in Table 1, similar to the observation in (Raghu et al., 2020), indicating the effectiveness of ANIL method. However, ANIL-HSML and ANIL-ARML perform similarly to ANIL-MAML, losing their superiority of modeling the mixture distribution of tasks achieved when implemented without ANIL as in Table 1 (up to 6.08% average improvement). This is because the cluster layers in HSML and the graph layers in ARML both affect the embeddings learned through backpropagation, i.e., they were designed for joint training with the encoder. When the encoder is frozen, they cannot work properly. For this reason, to be consistent with the existing works (Yao et al., 2019a, 2019b) that demonstrated the differences between HSML/ARML and MAML, we used their original designs in Table 1 and Table 2, and reported the ablation study with ANIL in Appendix D.5.
>
> Q3. Could you please discuss the limitation or in which case the algorithm will fail?
>
> In the case when the task distribution is not a mixture, our model would degenerate to and perform similarly to the generic metric-based meta-learning methods, e.g., ProtoNet, which considers a uni-component distribution. To confirm this, we added an experiment that compares our model with ProtoNet-Aug on Mini-Imagenet, which does not have the same explicit mixture distributions as in the Plain-Multi and Art-Multi datasets in Section 4. The results are summarized in the following table. From the table, we observe our method performs comparably to ProtoNet-Aug, which validates the aforementioned guess. Meanwhile, together with the results in Table 1 and 2 in the draft, the proposed method could be considered as a generalization of the metric-based methods to the mixture of task distributions. We added this discussion to Appendix D.6 in the revised draft.
>
> |Model       | 5-way-1-shot | 5-way-5-shot |
> |------------|--------------|--------------|
> |ProtoNet-Aug|59.40±0.93    |**74.68±0.45**|
> |HTGM        |**61.80±0.95**|74.55±0.45    |

---

> ### Author Response · Authors · 2022-12-12
> **Thank you and looking forward to hearing from you**
>
> Dear Reviewer,
>
> Thank you for your insightful comments. Following your valuable suggestions, we included more discussions on the related work of multi-task learning, and the difference between the proposed method and the multi-task learning methods. We also added new experimental results on the compared methods, and a discussion on the limitation of the proposed method. Your suggestions are very helpful for us to improve our work, and we would be very grateful if we could hear from you to see if our response helps address the questions, and take the opportunity before the end of the discussion period to address any further questions.
>
> We also added a summary of the revisions to the paper in a comment above for potential help with the review of the revised paper.
>
> Thank you for your attention!

---

### Author Response · Authors · 2022-11-19
**Appreciate your attention and time**

Dear Reviewers,

We sincerely appreciate your valuable comments that help us refine our work. If you have more questions or concerns about our response or current draft, please let us know. We are happy to discuss with you.

---

### Author Response · Authors · 2022-12-05
**Thanks for your reviews**

Dear reviewers,

Thank you for reviewing our paper and providing the constructive comments. As the discussion period is approaching to its end, we would appreciate if the reviewers could let us know if our response could address the questions and sufficiently justify our work. We would also appreciate if the reviewers could let us know where we can refine or what concerns you still have, so that we can take the opportunity to clarify them and improve our work. Thank you!

---

### Author Response · Authors · 2022-12-09
**A summary of the paper revision**

We thank all the reviewers for the insightful feedback and the suggestions for improving the paper. We are glad that the reviewers found our work studied a novel and important problem in meta-learning. We tried to address the questions and take the suggestions by adding appropriate discussions, performing additional experiments and revising the paper. Here please let us provide a summary of the major revisions to the paper for the possible help with the review of the revised paper.

1. **In the last paragraph of Section 2 (referred to Appendix B.2)**, we added a discussion on the related works of multi-task learning (following Reviewer **A867**'s suggestion).
2. **In the last paragraph of Section 2 (referred to Appendix B.1)**, we added a discussion on the related works of the conventional Hierarchical Gaussian Mixture Models (following Reviewer **DtsZ**'s suggestion).
3. **In Section 3 and Figure 1**, we revised the notations of the means and variances by using proper superscripts to improve the readability (following Reviewer **gaxY**'s suggestion in Q3).
4. **At the end of the "Task-Conditional Distribution" in Section 3.1 (referred to Appendix B.3)**, we added a discussion on the interpretation of the parameter W in Eq. (2) (following Reviewer **gaxY**'s suggestion).
5. **In Section 3.3**, we clarified the sample step of embeddings from the posterior distribution $q$ (following Reviewer **gaxY**'s suggestion).
6. **In Section 3 and Figure 1**, we revised the notation of the parameter of the inference network from $\psi$ to $\phi$, added $q_{\phi}$ at proper places in **Figure 1**, and added a sentence at **the fourth line in Section 3.3** to help recall the meaning of parameter $\phi$ (following Reviewer **gaxY**'s suggestion).
7. **In Section 4, above Section 4.1 (referred to Appendix D.5)**, we added the experiments on the comparison with meta-learning baseline methods that use the same backbone networks as the proposed model, in a way following the ANIL method (following Reviewer **A867** and **DtsZ**'s suggestions).
8. **In Section 4.1, "novel task detection" subsection, the second paragraph (referred to Appendix C.2)**, we added a description on the setup of the experiments on novel task detection (following Reviewer **gaxY**'s suggestion).
9. **In Section C.1**, we elaborated the configuration setup and the selection method of hyperparameters for each of the compared methods (following Reviewer **gaxY**'s suggestion).
10. **In Appendix D.6**, we added experiments for a discussion on the limitation of the proposed method when the task distribution is not a mixture distribution (following Reviewer **A867**'s suggestion).

We are happy to take the opportunity to address any further questions, and we also look forward to the discussion with the reviewers. Thank a lot!

---

### Decision · Program_Chairs · 2023-01-20

**Decision:**

Reject

**Justification For Why Not Higher Score:**

N/A

**Justification For Why Not Lower Score:**

N/A

**Metareview: Summary, Strengths And Weaknesses:**

This paper proposed a new meta learning framework based on a Hierarchical Gaussian Mixture based Task Generative Model (HTGM) to address the issues in meta-learning: (1) the various sources of tasks may compose a multi-component mixture distribution, and (2) novel tasks may come from a distribution that is unseen during meta-training. The paper is well written, the proposed idea is well motivated, and the work is overall well executed. However, as pointed by the reviewers, the novelty of the proposed technique (most technical contributions are there in literature) is limited, and the empirical studies are not convincing enough (results on the selected dataset do seem strong but there is no evaluation on larger-scale general-domain image datasets such as Mini-ImageNet, etc. Overall, this is a well-written paper but the technical merits and empirical results are slightly below the acceptance bar.